# Do streamwater solute concentrations reflect when connectivity occurs in a small pre-alpine headwater catchment?

Leonie Kiewiet[1], Ilja van Meerveld[1], Manfred Stähli[2], Jan Seibert[1,3]

[1]Department of Geography, University of Zürich, Zürich, Switzerland
[2]Swiss Federal Research Institute WSL, Birmensdorf, Switzerland
[3]Department of Aquatic Sciences and Assessment, Swedish University of Agricultural Sciences, Uppsala, Sweden

*Correspondence to*: Leonie Kiewiet (leonie.kiewiet@geo.uzh.ch)

**Abstract.** Expansion of the hydrologically connected area during rainfall events causes previously disconnected areas to contribute to streamflow. If these newly contributing areas have a different hydrochemical composition than the previously
connected contributing areas, this may cause a change in streamwater chemistry that can not be explained by simple mixing of rainfall and baseflow. Changes in stormflow composition are, therefore, sometimes used to identify when transiently connected areas (or water sources) contribute to stormflow. We identified the dominant sources of streamflow for a steep 20-ha pre-alpine headwater catchment in Switzerland and investigated the temporal changes in connectivity for four rainfall events based on streamwater concentrations and groundwater level data. First, we compared the isotopic and chemical composition
of stormflow at the catchment outlet to the composition of rainfall, groundwater, and soil water. Three-component end-member mixing analyses indicated that groundwater dominated stormflow during all events, and that soil water fractions were minimal for three of the four events. However, the large variability in soil and groundwater composition compared to the temporal changes in stormflow composition inhibited the determination of the contributions from the different groundwater sources. Second, we estimated the concentrations of different solutes in stormflow based on the mixing fractions derived from two-
component hydrograph separation using a conservative tracer ($\delta^2$H) and the measured concentration of the solutes in baseflow and rainfall. The estimated concentrations differed from the measured stormflow concentrations for many solutes and samples. The deviations increased gradually with streamflow for some solutes (e.g., iron and copper), suggesting increased contributions from riparian and hillslope groundwater with higher concentrations of these solutes, and thus increased hydrological connectivity. The findings of this study show that solute concentrations partly reflect the gradual changes in hydrologic
connectivity and that it is important to quantify the variability in the composition of different source areas.

## 1 Introduction

During dry periods only a small part of a catchment is connected to the stream, but the connected area can expand dramatically during rainfall or snowmelt events (Stieglitz et al., 2003; Bracken and Croke, 2007; Jencso and McGlynn, 2011; van Meerveld et al., 2015). Knowledge of which areas are connected and contribute to streamflow is important because it helps us to shape
our conceptual understanding of how catchments function. For example, Ladouche et al. (2001) showed for the 0.8 km[2] Strengbach catchment in France that the upper layers of saturated areas, contributed up to 30% of the discharge during the initial stages of a rainfall event, even though these areas occupied only 2% of the catchment area. However, during the final stage of an event, upslope and downslope areas contributed equally to flow. Similarly, Oswald et al. (2011), showed for a 0.8 km[2] catchment in north-western Ontario, Canada, that a large part of the catchment area was hydrologically disconnected from
the stream during most events, and that there was a threshold catchment storage at which a larger area contributed to streamflow. Connection of upslope areas does not only lead to large changes in discharge (Lehmann et al., 2007; Detty and McGuire, 2010; van Meerveld et al., 2015) but can also cause major changes in streamwater composition (e.g., Devito and Hill, 1997; Stieglitz et al., 2003; Ocampo et al., 2006). Interpretations of hydrologic connectivity are often based on such changes in streamwater chemistry (Uhlenbrook et al., 2004; Soulsby et al., 2007; Pacific et al., 2010).


Hydrologic connectivity, i.e., "the linkage of separate regions of a catchment via water flow" (Blume and van Meerveld 2015) is usually inferred from either stream-based or hillslope-based measurements, because direct observations of connectivity are limited due to the difficulty in observing and quantifying subsurface processes (Hopp and McDonnell, 2009; Blume and van Meerveld, 2015). In many studies, conservative tracers (e.g. stable water isotopes or non-reactive elements) are selected to identify the origin of streamflow, using methods such as hydrograph separation (Buttle, 1994) or end-member mixing analyses (EMMA; Hooper et al., 1990; Christophersen and Hooper, 1992). Tracers can also be used to assess connectivity of hillslopes to the streams (Tezlaff et al., 2014; Uhlenbrook et al., 2004). Since stream chemistry is the proportional mixture of all actively contributing areas, quantifying each contribution results in a measure for catchment-wide connectivity. For instance, McGlynn and McDonnell (2003) used silica concentrations and isotope data for a 2.6-ha sub-catchment of the Maimai catchment in New Zealand to show that the contributions from the hillslopes were larger for an event with higher wetness conditions than for an event with drier initial conditions and were also larger on the falling limb of the hydrograph. Several studies in the 31 km$^2$ Girnock Burn catchment in Scotland investigated connectivity of source areas to the stream using Gran alkalinity and isotope data (e.g., Soulsby et al., 2007; Tezlaff et al., 2014). They found that the upper soil layers and upslope areas increasingly dominated streamflow at higher flows and that the riparian peat soils modulated the streamwater isotopic composition. However, few studies have compared the results from stream-based and hillslope-based inferences of connectivity. Burns et al. (1998) showed that hillslope contributions to streamflow inferred from end-member mixing analysis were similar to the subsurface flow measurements for a trenched hillslope.

Mixing analyses are traditionally performed with conservative solutes and stable water isotopes (Hooper and Shoemaker, 1986). Non-conservative solute concentrations can also provide useful information on hydrological connectivity and flow pathways because they can aid the identification of different source areas (Barthold et al., 2011; Abbott et al., 2016). The concentrations of specific elements can also be indicative for differences in redox conditions (e.g., sulfate, iron, manganese), bedrock-contact time (e.g., calcium, magnesium, sodium, barium) or vegetation (e.g., nitrogen, phosphorus, potassium) (Kaushal et al., 2018). It has been suggested that the discrepancy between hydrograph separation results for conservative and non-conservative tracers highlights when and where streamwater is not the result of conservative mixing between end-members, such as baseflow and precipitation (Kirchner, 2003). Instead, it might reflect mixing from different 'old' water sources in the catchment that have different concentrations. Therefore, this discrepancy may provide information on when hillslope-stream connectivity is established. Alternatively, the differences in the relative response of conservative and non-conservative tracers during rainfall events might be (partly) due to reactive processes that mobilize (or immobilize) solutes at the event time-scale (Godsey et al., 2009). As such, focusing on solute responses in stormflow and the difference between conservative and non-conservative tracers might allow us to identify the extent of these reactive transport processes and contributions from 'old' water sources that do not contribute to baseflow.

Solute concentrations in streamwater might be relatively constant (chemostatic), decrease (dilution) or increase (mobilization) in response to rainfall, depending on the source areas to streamflow and their respective concentrations, as well as reactive transport processes (Godsey et al., 2009; Seibert et al., 2009, Knapp et al., 2020). Godsey et al. (2009) found that concentrations of typical weathering products (calcium, magnesium, silica and sodium) were nearly chemostatic for 59 geochemically diverse US catchments, suggesting a (constant) source of these solutes. This implies that the areas that contribute to streamflow during rainfall events have similar concentrations of these solutes as the permanently contributing areas, higher concentrations to compensate for the dilution caused by the rainfall, or that reactions are fast enough to maintain similar concentrations during the event.

The timing of the onset of contributions from different source areas also affects the solute concentrations (Abbott et al., 2018). Several studies have shown that the relationship between concentrations and discharge is hysteretic at the event time-scale (e.g., Evans and Davies, 1998; Hornberger et al., 2001). Zuecco et al. (2019) showed that the increase in subsurface connectivity was delayed compared to streamflow (anti-clockwise hysteresis) for two sub-catchments of the Studibach catchment in Switzerland, suggesting that hillslope runoff may not be the dominant runoff source at the beginning of rainfall events for these small catchments. If hillslope and riparian zone water have a different composition, this can cause hysteresis in the relation between solute concentrations and streamflow. Changes in solute concentrations might also depend on the size of the catchment (Brown et al., 1999) and mixing that occurs during transport from the source areas to the outlet. For instance, hillslope runoff may bypass the riparian zone through focused locations along the stream channel or via preferential flow pathways (Allaire et al., 2015), and mix with other hillslope sources (Seibert et al., 2009) and riparian groundwater (McGlynn and McDonnell, 2003; Chanat and Hornberger 2003) on its way to the stream.

For all analyses of source areas and connectivity it is important to quantify the variability in the concentrations of conservative and non-conservative tracers because it affects the robustness of the results and thus interpretations of connectivity. However, for most small catchment studies it remains unclear how large the changes in streamwater composition are compared to the spatial variability in groundwater and soil water because the spatial variability in groundwater and soil water are rarely assessed (<10 km$^2$; Penna and van Meerveld, 2019). In this study, we combined spatially distributed soil- and groundwater sampling with event-based streamwater sampling in the pre-alpine Studibach catchment to address the following research questions:

1. How variable is streamwater chemistry during events compared to the spatial variability in soil and groundwater chemistry?
2. What are the dominant sources of streamflow during small to intermediately sized rainfall events?
3. How much do the changes in the concentrations of conservative and non-conservative tracers differ during events and does this difference provide information on the relative contributions of different parts of the catchment and, thus, hydrological connectivity?

## 2. Study catchment

We conducted this study in the 0.2 km$^2$ pre-alpine Studibach catchment, a headwater catchment of the Zwäckentobel, located in the Alptal, canton Schwyz, Switzerland. The elevation of the Studibach ranges from 1,270 to 1,650 m above sea level. The mean annual precipitation is about 2,300 mm y$^{-1}$. The precipitation is relatively evenly distributed throughout the year (Feyen et al., 1999) and about one-third falls as snow (Stähli and Gustafsson, 2006). The catchment is steep (average slope: 35°) and characterized by a step-wise topography, with flatter areas and steep slopes due to soil creep and landslides. An open coniferous forest covers about half of the catchment (Hagedorn et al., 2000), a third is characterized as a moor landscape or wet grassland, and the remaining areas are alpine meadows.

Streamflow and groundwater levels respond quickly to rainfall (Fischer et al., 2015; Rinderer et al., 2015). The groundwater level response time is generally less than 30 minutes (Rinderer et al., 2014) and only 3-mm of cumulative rainfall already causes an increase in the groundwater level for a large part of the catchment during typical conditions (Rinderer et al. 2015). The groundwater level peak precedes the peak discharge in the Studibach at half of the sites, but only by 15 or 20 minutes (Rinderer et al., 2015). Water levels in flatter locations and topographic depressions rise nearly instantaneously, which suggests that they can contribute to streamflow during the early stages of a rainfall event. Previous studies suggest that event water fractions in stormflow are generally low (Kiewiet et al., in press; von Freyberg et al., 2018), except for events with more than 50-mm of rainfall (Fischer et al., 2017).

Soils are generally shallow (0.5 m at ridge sites to ~2.5 m in depressions); soil depth is weakly correlated to slope (van Meerveld et al., 2018). The gleysols are underlain by three different types of Flysch bedrock, which is a reworked carbonate rock consisting of deep-water deposits. The carbonate-rich bedrock results in high groundwater concentrations with a calcium-bicarbonate signature, although some sites have high sulfate and magnesium concentrations (Kiewiet et al., 2019).

The Studibach can be subdivided into four different landscape elements with a distinct groundwater composition (Kiewiet et al., 2019 and Fig. 1):

1.  Riparian zone, flatter areas and topographic hollows with above-average concentrations of iron and manganese. These areas are from here on referred to as 'riparian';
2.  Hillslopes and steeper areas, characterized by above-average concentrations of copper, zinc and lead;
3.  Areas with above-average concentrations of weathering-derived solutes, such as strontium, indicative of longer (and deeper) flow pathways, which are from here on referred to as deep groundwater;
4.  Areas located in a specific part of the catchment that is characterized by high magnesium and sulfate concentrations.

## 3. Methods

### 3.1 Hydrometric measurements

To monitor streamwater and groundwater levels, we used a network of 51 shallow groundwater wells and streamflow gauges (Fig.1) that was installed in 2009−2010 (Rinderer et al., 2014). The wells were distributed based on the topographic wetness index (TWI, Beven and Kirkby (1979)) and cover the range of wet and dry locations in the catchment. All wells were drilled by hand to the bedrock (0.5 to 2.5 m depth), screened over the entire length, except for the top ten centimeters, and sealed with a layer of bentonite clay. Stream stage was measured directly in the stream (outlet; Fig. 1a) or behind a V-notch weir (C5). Water levels were measured at each well and stream location with either a capacitance water level logger (Odyssey Dataflow Systems Pty Limited) or a pressure transducer (DCX-22 CTD Keller AG für Druckmesstechnick or STS DL/N 70, Sensor Technick Sirnach AG). The pressure data were corrected for changes in barometric pressure and temperature using the data from the MeteoSwiss station in Einsiedeln (910 m a.s.l; ca. 10 km from the catchment outlet). Rainfall was recorded at three locations within the catchment with tipping bucket rain gauges (0.2 mm resolution, Odyssey Dataflow Systems Pty Limited; Fig.1a).

The stream stage data were converted to specific discharge (Q, further referred to as discharge) using a rating curve based on twenty salt dilution measurements. Due to technical issues, there were no observations of stage height at the catchment outlet during events I and II (see section 3.2). We used the correlation between the specific discharge at the catchment outlet and an intermediately sized sub-catchment (C5, Fig.1a) for the four months following events I and II to estimate the streamflow at the outlet for the period without data (coefficient of determination $r^2$ = 0.66, RMSE = 0.75 mm h$^{-1}$, for comparison the 10$^{th}$ and 90$^{th}$ percentile of Q at the catchment outlet for this period were 0.35 and 2.11 mm h$^{-1}$, respectively). We assume that the uncertainty in the discharge for events I and II does not affect our conclusions as they are largely based on relative changes in streamflow during the events. The ranking of the events based on the peak of the (reconstructed) discharge was the same as the ranking based on the peak rainfall intensity.

### 3.2 Sample collection

We analysed streamflow and stream chemistry for four events (I-IV; Table 1) in the fall seasons of 2016 and 2017. Stream water samples were collected at the outlet of the Studibach using automatic samplers (full-size portable sampler, 3712, ISCO

Teledyne, USA). The sampling interval was based on the expected event duration. The multi-interval program was set to
sample streamwater every ten to twenty minutes at the start of the rising limb (maximum of six samples). The remaining
eighteen samples were taken at an hourly-interval. We emptied the samplers within 24 hours after sample collection to avoid
fractionation. We used a timer to start the sampler if the expected time of the onset of the rainfall was during the night. Rainfall
was collected with passive sequential samplers (built after Kennedy et al. (1979), and described in detail in Fischer et al.
(2019)) at two locations in the catchment (rain gauge location one and two in Fig. 1a). The samplers collected a sample for
approximately every 5 mm of rainfall.

For soil water and groundwater, we used the data from a subset of nine baseflow snapshot campaigns during the snow-free
seasons of 2016 and 2017 (Kiewiet et al., 2019). Soil water was collected with six to 18 suction lysimeters at four to six sites
(at 15, 30 and 50 cm below the surface at forested and non-forested sites at three different elevations: 1361, 1502, 1611 m
a.s.l.; Fig.1a). We applied a tension of 50 mbar to the lysimeters and collected the soil water sample the next day. Groundwater
was collected at all wells that contained water (34 to 38 wells). The shallow wells were either purged or at least twice the well
volume was extracted a day before the sampling. For a detailed description of the groundwater sampling procedure, see Kiewiet
et al. (2019).

### 3.3 Sample analyses

The samples for cation and anion analyses were stored in a fridge (6 °C) before lab analyses (within a few days) or were frozen
(-18 °C) directly after collection until shortly before the analyses. The samples were filtered (0.45 µm; SimplepureTM Syringe
Filter) and acidified (only for cation analysis) to mobilize trace metals. The samples were analysed at the Physics of
Environmental Systems laboratory at ETH Zurich (Switzerland) using an ion-chromatograph (861 Advanced Compact IC,
Metrohm) for anions and a mass-spectrometer (ICP-MS 9700, Agilent technologies) for cations. Calibration curves were
obtained from measurements with five calibration standards before or after measuring the samples.

The samples were analysed for stable water isotope composition with a cavity ring-down spectroscope (L2140-I (CRDS) or
L2130-I (CRDS), Picarro, Inc., USA) at the Chairs of Hydrology at the University of Freiburg (Germany). The reported
precision is ± 0.16 ‰ for $\delta^{18}O$ and ± 0.6 ‰ for $\delta^{2}H$. All samples plotted close to the local meteoric water line. The average (±
standard deviation) of the Line Conditioned-excess (LC-excess; Landwehr and Coplen (2006)) for all 516 stream-, soil- and
groundwater samples was 5.3 ± 1.3 ‰, excluding five soil water samples (taken at 15 (three samples), 30 (one sample) and 50
(one sample) cm below the soil surface) for which LC-excess ranged from –9.6 to -1.5 ‰. Deuterium-excess ($D_{ex}$) was
calculated as $D_{ex} = \delta^{2}H - (8 \cdot \delta^{18}O)$.

### 3.4 Data analysis

### 3.4.1 Relative concentrations

We examined the changes in streamwater concentrations during the rainfall events using concentration-discharge (C-Q)
relationships and identified the corresponding hysteresis index (cf. Zuecco et al., 2016). For this, we normalized both the
discharge and the concentrations so that zero represents the smallest measured value, and one the highest measured value.

For each solute, we calculated the relative concentration $R_x$ by comparing the concentration of the sample to that of baseflow:

$$R_x = \frac{c_{Q\_x}}{c_{BF\_x}} \qquad (2)$$

Where $C_{Q\_x}$ and $C_{BF\_x}$ are the concentration of solute $x$ in streamwater during the event and in baseflow before the event. We
define baseflow as the streamflow between rainfall-runoff events and assume that it comes from groundwater. The relative

concentration indicates dilution ($R_x < 1$) or enrichment ($R_x \geq 1$)) during the events. It thus quantifies the direction and magnitude of the change in solute concentrations (note that $R_x$ is not an alternative measure for the fraction of baseflow in stormflow). We used the relative concentrations ($R_x$, Eq. 2) to identify groups of solutes using hierarchical cluster analysis.

### 3.4.2 Hydrograph Separation and End-Member Mixing Analysis

We tested if the median concentrations of different (ground)water types were significantly different (Table 2; Tukey-Kramer test; Tukey.HSD in the 'agricolae' R-package). We pairwise tested seven groups: all groundwater, riparian groundwater, hillslope groundwater, all soil water, soil water at forested sites, soil water at non-forested sites, and rainfall. We performed all computations in R (R core team, 2013) and used a 95-percent confidence interval for all statistical tests. We found that the soil water samples taken at forested or non-forested sites were never significantly different, and thus merged these data.

We investigated the sources of streamflow using two and three-component mixing analyses and investigated the difference between the observed solute concentrations and those estimated assuming linear mixing of baseflow and rainfall. Ideally, we would use the soil water and groundwater samples taken directly before the rainfall events, but these data are not available. Instead, we have data from sampling campaigns two to nine days before (event II) or after the events (I, III and IV). Since the spatial variability in groundwater composition in the Studibach is larger than the temporal variability (Kiewiet et al., 2019), we assume that the groundwater and soil water samples reflect the typical composition and variability of soil water and groundwater, but acknowledge that absolute concentrations might have been slightly different. A Principal Component Analysis (PCA) on the chemical and isotopic composition of all groundwater (n=335) and soil water (n=116) samples (z-transformed) showed that soil water and groundwater were consistently different in the principal component space; only six of the soil water samples (5%) plotted within the same area as the groundwater samples (see S1 for the PCA result and Table 2 for the average concentrations).

We estimated the fraction of event ($f_e$) and pre-event ($f_{pe}$) water in the streamwater samples ($C_t$) using two-component isotope hydrograph separation (Eq. 1). The results for $\delta^2H$ and $\delta^{18}O$ were similar (difference between the event-average $f_{pe} \leq 0.05$). Because the ratio of precision to range was better for $\delta^2H$, we report only the $\delta^2H$ results. A pre-event baseflow sample was used to characterize the pre-event water composition ($C_{pe}$). The incremental weighted mean of rainfall was used to characterize the event-water composition ($C_e$).

$$f_{pe} = \frac{C_t - C_e}{C_{pe} - C_e} \qquad (1)$$

We also estimated the fractions of groundwater, soil water and rainwater in each streamwater sample, using a three-component End Member Mixing Analysis (EMMA; Christophersen and Hooper (1992)). We based the EMMA on the first two principal components of a PCA that included all conservative tracers. We considered a tracer conservative if the concentration was linearly correlated to that of at least one other tracer (cf. Barthold et al., 2011). To determine the conservativeness, we used all groundwater, soil water and streamwater samples used in this study (n=549), and set the threshold for a linear correlation to $R^2 \geq 0.5$ and $p < 0.01$. EC, calcium, magnesium, barium, $\delta^2H$ and $\delta^{18}O$ were conservative based on this definition; the other tracers (e.g., copper, sulfate, potassium, and iron) were not. However, note that this threshold does not per se imply a linear trend and that although a linear trend is consistent with conservative mixing, it does not necessarily confirm conservative mixing either (James and Roulet, 2006).

We used a Gaussian error-propagation method (Genereux, 1998) to estimate the uncertainty in the calculated fractions of the source waters for the two-component hydrograph separation and EMMA. For the two-component hydrograph separation, we

defined the uncertainty in the event and pre-event water composition as the standard deviation of the rainfall sampled during the event, and groundwater sampled during the snapshot campaign closest to the event (see Table 1), respectively. For the uncertainty in the EMMA, we used the standard deviation of groundwater, soil water and rainwater samples for the event. We used the laboratory accuracy for the uncertainty of the streamwater samples in the two-component hydrograph separation, and for the EMMA assumed that the uncertainty for the streamwater samples in the principal component space was similar to the standard deviation of the last three streamwater samples taken during each event (i.e. the last streamflow samples taken at the falling limb of the hydrograph). We multiplied the standard deviation with a t-value based on the number of samples and used a 95-percent confidence interval for all uncertainty estimations.

### 3.4.3 Deviation of concentrations from mixing of baseflow and rainfall

We compared the measured streamflow concentrations for each solute to the concentration that would be expected based on conservative mixing of rainfall and baseflow ($C_{es}$):

$$C_{es\_x} = \left(C_{BF\_x} \cdot f_{pe}\right) + \left(C_{P\_x} \cdot (1 - f_{pe})\right) \quad (3)$$

where $C_{es\_x}$ is the 'estimated' concentration for solute $x$, $C_{BF\_x}$ and $C_{P\_x}$ are the concentrations for solute $x$ in baseflow and rainfall (average rainfall composition: Table 2), and $f_{pe}$ is the pre-event water fraction for that sample, as determined from the two-component hydrograph separation using $\delta^2 H$ as the tracer (Eq. 1).

We compared the estimated ($C_{es\_x}$) and measured streamflow ($C_{Q\_x}$) concentrations for each sample and solute to assess the relationship between discharge and the potential contribution of different source areas. We assumed that overestimation of the concentrations ($C_{Q\_x}/C_{es\_x} > 1$) indicates either a contribution from source areas that have a higher concentration than the sources that contributed to baseflow, or reactive transport. Similarly, underestimation of the concentrations ($C_{Q\_x}/C_{es\_x} < 1$) indicates either a contribution from source areas that did not contribute during baseflow and have a lower concentration than the sources that contributed to baseflow, or reactive transport. Given the characteristic concentrations in different (ground)water types (Table 2 and 3, Fig. 2), we interpret the changes in the streamwater composition during an event as following: 1) higher copper and nickel concentrations are indicative of flow from hillslopes and forested areas, 2) higher iron and manganese concentrations are indicative of flow from riparian areas, 3) higher $D_{ex}$, barium, magnesium concentrations are indicative of soil water, 4) higher potassium concentrations can indicate either soil water or hillslopes groundwater. However, note that the variability for soil water, groundwater and rainfall was large (Table 2, and see supplement S2 for boxplots of tracer concentrations in each water compartment). Also, the non-conservative nature of these tracers should be taken into account. For instance, iron and manganese are mainly soluble under anoxic, reducing conditions, such as in the riparian areas, but they might oxidize and form an insoluble compound after entering the streams. Adsorption of metals (e.g., iron, copper, zinc) to organic compounds or clay particles may also influence the concentrations in streamflow, and their concentration may be underestimated if they are adsorbed to coarser particles that settle out during streamflow recession (Kaushal et al., 2018). The concentration of some solutes is, furthermore, controlled by weathering processes or influenced by plant-uptake because they are macro (potassium, magnesium) or micro (e.g., copper, nickel) plant nutrients. In this study, we assume that concentration increases or decreases due to weathering or plant-uptake are negligible at the event (i.e., hourly) time-scale.

### 3.4.4 Groundwater-level-based connectivity assessment

We investigated in how far stream chemistry reflects conservative mixing of baseflow and precipitation and whether this breaks down at a certain specific discharge or reflects an increase in hydrologic connectivity. We related the ratio of the

estimated and measured concentrations ($C_{Q\_x}/C_{es\_x}$, see 3.4.3) for each solute to the discharge and the calculated fraction of the cachment that was connected to the stream. We used the data-driven model of Rinderer et al. (2019) to determine which parts of the catchment were active and connected to the stream. This model uses the water level data from all 51 wells in the catchment and time series clustering to assign each pixel in the catchment to one of six groundwater level clusters. For each time step, the average relative groundwater level for all monitoring wells that belong to a cluster is calculated and assigned to all pixels in that cluster. This relative water level is then transformed into an absolute water level based on the correlation between soil depth and slope. If this simulated water level is within 30 cm of the soil surface (i.e., the part of the soil where the hydraulic conductivity is high), the pixel is considered active, otherwise, it is considered inactive. If a pixel is active and, based on surface topography, connected to the stream via other active pixels, it is assumed to be connected to the stream. We thus assume that significant lateral flow occurs when the water table rises into the near-surface layers, where the hydraulic conductivity is much larger (cf. Schneider et al., 2014). Hence, the simulated connectivity refers to the connectivity of groundwater flow in the more permeable layer of the soil above the more permanently saturated soil. In the Studibach, there is an almost permanent water table in the low conductivity gleysols in most locations. It is thus not so likely that the lateral water flow would infiltrate into the bedrock before reaching the stream (Jackson et al., 2014). Rinderer et al. (2019) tested the sensitivity of the method for misclassification of the clusters by randomly re-assigning pixels to different clusters and the uncertainty in the soil depth by comparing the connectivity time series to the time series computed with a different (DEM-based) soil depth map. The soil depth had only a minor influence on the model results (RSME > 0.0003% of the relative soil depth). Still, misclassification of pixels (i.e., assigning them to a different cluster) could result in up to an 8% difference in the simulated connected area between the different model runs.

## 4. Results

### 4.1 Event characteristics

Total rainfall for the four events ranged between 17 and 33 mm (Table 1, Fig. 3). The duration of the events ranged from 7 to 27 hours. The four events were larger than the long-term average daily precipitation and within the upper 30[th] percentile of daily precipitation at the long-term meteorological station Erlenhöhe, located 500 meters from the catchment outlet (median: 10.0 mm; mean ± sd: 14.1 ± 13.8 mm for all 7452 days with more than 1 mm of precipitation between 1981-2017; Stähli, 2018). However, the events were smaller than the 50 mm threshold for large contributions of event-water to streamflow (Fischer et al. 2017). The average and maximum 10-minute rainfall intensities ranged between 1.2 and 3.9 mm h$^{-1}$ and between 4.8 and 22.8 mm h$^{-1}$, respectively.

Discharge at the catchment outlet increased the least (from 0.02 to 0.07 mm h$^{-1}$) for the smallest event (I), and most for event III (0.08 to 0.43 mm h$^{-1}$). The simulated fraction of the catchment that was hydrologically connected to the stream varied from 0.27 (before the start of events I and II) to 0.68 (at the time of peak flow for event III) (Fig. 4). The relation between the simulated fraction of the catchment that was connected to the stream and discharge was non-linear for all events (Fig. 5, top row). For all of the four events, connectivity was lower on the rising limb of the hydrograph than on the falling limb for the same discharge. For event I, the connected area increased significantly at the recession of the streamflow. For event II connectivity increased little during the sampling period (0.27 to 0.28). Discharge increased to >4 mm h$^{-1}$ after the sampling period of event II due to additional rainfall, but interestingly the simulated connectivity increased only marginally (up to 0.35; see S3) during this period. During the smaller events with initially low connectivity, the hydrologically connected area extended laterally from the stream up, but remained confined to the flat areas. For the intermediate events (III and IV), the lateral extension was larger, and parts of the hillslopes became connected. However, the data-based model suggested that

during all four events, large parts of the catchment remained hydrologically disconnected from the stream network (Table 1, Fig. 4).

## 4.2 Concentration-discharge relationships

The chemical and isotopic composition of streamwater changed during all four events, but the magnitude and direction of the response differed for each event and solute (Fig. 5). Hysteresis in the relation between solute concentrations and discharge depended on the event size and differed between solutes (Table 3, Fig. 5). During events III and IV, the relation between discharge and concentration was hysteretic for most solutes. The double discharge peaks during events I and II (Fig. 2) resulted in a double loop in the concentration discharge relationship for deuterium, iron, and calcium (Fig. 5).


The average relative concentration (average $R_x$ for the streamflow samples taken during the four events, n=100, Eq. 2) for deuterium excess ($D_{ex}$) and chloride were 4.1 and 2.0, respectively. This reflects the substantial increase in these concentrations during events. Manganese and iron concentrations also increased with increasing discharge, but less than $D_{ex}$ and chloride (mean $R_x$: 1.0 for both iron and manganese; maximum $R_x$: 2.8 and 3.2, respectively). On average, the concentrations of copper,

nickel and zinc decreased with increasing discharge (mean $R_x$: 0.78, 0.63 and 0.31), but individual stormflow samples were enriched up to 1.7, 1.3 and 1.1 times the baseflow concentration, respectively. Concentrations of iron and copper were higher on the falling limb than on the rising limb (counter-clockwise hysteresis). Event I was the only event during which copper concentrations did not increase with increasing discharge.

The concentrations of sodium, magnesium, calcium and barium decreased with increasing discharge (mean $R_x$: < 0.77). The concentrations of these solutes, and also sulfate, were higher on the rising limb than on the falling limb (resulting in clockwise hysteresis). Sulfate concentrations decreased with increasing discharge during events I, III and IV but increased with discharge during event II. Potassium and sulfate concentrations (range $R_x$: 0.2−1.7 and 0.3−1.4, respectively) were highest shortly after the onset of an event (first four samples) and decreased afterwards. These differences in the magnitude and timing of the

change in solute concentrations and isotopic composition allowed for subdivision of the tracers into different groups based on the computed $R_x$ values for all events (A to D; Table 3, Fig. 6).

## 4.3 Hydrograph separation and End Member Mixing Analysis results

Two-component hydrograph separation indicated that most stormflow was 'old' water (Fig. 3; Table 3). The maximum event water fraction ($f_e$) was highest for event II ($f_e = 0.24\pm0.61$) and smallest for event IV ($f_e = 0.14\pm0.28$). However, the differences

between the events were much smaller than the associated uncertainties (Table 4). The high event water fraction of event II occurred when the connected area was relatively small. The fraction of connected area during event II expanded only 0.01 (up to 0.28) during the period that we sampled (see S3).

It was possible to calculate the relative fractions of groundwater, soil water and rainwater in stormflow for all events based on

EMMA as well (Table 4). Groundwater dominated streamflow during all events (range $f_{GW}$: 0.49±0.14 to 0.81±0.19). The event-average soil water fraction was considerable during event II ($f_{SW}$: 0.27), but negligible during the other events ($f_{SW}$: ~0). The event-average pre-event water fractions based on the EMMA (i.e., the sum of the groundwater and soil water fractions) were similar to the pre-event water fractions estimated using $\delta^2H$ as a tracer in the two-component hydrograph separations (range $f_{GW} + f_{SW}$: 0.73 to 0.81 vs range $f_{pe}$: 0.76 to 0.86). Although the results were similar, the uncertainties for EMMA were

smaller than for the two-component hydrograph separation. The uncertainties for the EMMA results were mainly caused by the uncertainty in the groundwater fraction (contribution of the groundwater uncertainty to the total uncertainty: 97%, 50%, 94%, and 94% for events I-IV, respectively). This is due to the large contribution of groundwater to streamflow and the large

spatial variability in the groundwater composition. For event II, the uncertainty due to the soil water contributions was larger than for the other events (25% for event II vs. 0.01%, 3% and 5% for event I, III and IV, respectively).

The explanatory power of the first two principal components for all stormflow, soil water and groundwater samples was 76.3% for event I (PC1: 53.1%; PC2: 23.2%) and 82.0% for event III (PC1: 56.2%; PC2: 25.8%; Fig. 7a and c). For event II and IV the explanatory power was 72.6% and 83.8%, respectively (see S4). The most striking aspect of the mixing plots, however, is the small change in the composition of stormflow compared to the spatial variation in the composition of the soil and groundwater end-members (Fig. 7b and d). The observed changes in solute concentrations in streamflow were largest during event II (e.g., changes of 23 $\mu$gL$^{-1}$ for Ba; 39 mgL$^{-1}$ for Ca and 11 ‰ for $\delta^2$H) but this change was similar to or smaller than the standard deviation of the concentrations for the groundwater samples or soil water samples taken during the corresponding snapshot campaign (e.g., groundwater: 44 $\mu$gL$^{-1}$ for Ba, 27 mgL$^{-1}$ for Ca and 5.9 ‰ for $\delta^2$H; soil water: 22310 $\mu$gL$^{-1}$ for Ba, 23 mgL$^{-1}$ for Ca and 10.4 ‰ for $\delta^2$H; see Figure S2 for boxplots of the concentrations for the different water types).

## 4.4 Estimated solute concentrations based on conservative mixing of rainfall and baseflow

The concentrations estimated based on the assumption of conservative mixing between rainfall and baseflow ($C_{es}$, Eq. 3) differed from the measured stormflow concentrations ($C_Q$) for almost all solutes (Fig. 8). The measured concentrations for geogenic solutes (shown for calcium and sodium in Fig. 8) were lower than the estimated concentrations. The measured concentrations of sulfate were lower than estimated based on conservative mixing as well, except for event II. For potassium there was no clear pattern: the concentrations were underestimated and overestimated at both low and high discharge (Fig. 8). The measured concentrations of cobalt, copper, nickel and iron (solute groups A and C, see Fig. 6) were slightly lower than the estimated concentrations for low discharge, but (much) higher during high discharge (Fig. 8). There was no distinct threshold in the relation between $C_Q/C_{es}$ and either discharge or the simulated fraction of the catchment that was connected to the stream (Fig. 8 and S5), $C_Q/C_{es}$ rather changed gradually with increasing discharge and connected area.

## 5. Discussion

### 5.1 Small changes in streamflow composition compared to the spatial variability in groundwater and soil water

Changes in solute concentrations in streamwater during rainfall events depend on the changes in the relative contributions of different sources to streamflow (e.g., event and pre-event water, or different pre-event water sources), the differences in the concentrations of these sources, as well as reactive transport processes. Our results show that the change in streamflow composition during the four rainfall events was much smaller than the spatial variability in groundwater and soil water composition. For instance, the average change in the concentration of barium and deuterium in streamflow for the four events was similar to the spatial variability in shallow groundwater and soil water measured after events I and II (13.8 $\mu$gL$^{-1}$ Ba and 6.1 ‰ change in streamwater, versus an interquartile range of 30 $\mu$gL$^{-1}$ and 4.8 ‰ for shallow groundwater and 10.6 mgL$^{-1}$ and 5.7 ‰ in soil water). This was also evident from the principal component analysis and mixing plots (Fig. 7). It is to be expected that the change in streamwater composition is less than the variability between the end-members, but for a viable hydrograph separation, the change in streamwater composition should be larger than the variability within the end-members (Hooper, 2001). The change in streamwater composition during the four events presented in this study was not large enough to distinguish contributions from the different groundwater sources, although it is evident that pre-event water dominated streamflow.

We could show that the spatial variation within different source areas was large compared to the temporal variation because we collected a large dataset of groundwater and soil water samples. However, in other small catchment studies, this comparison

is often restricted because of insufficient spatial sampling (Penna and van Meerveld, 2019). Based on our experience for the Studibach, we see a clear need for further spatial sampling of groundwater and soil water in other catchments to determine this spatial variability.

**5.2 Which areas or sources contribute to stormflow?**

For the events included in this study, the estimated area that was hydrologically connected to the stream was never smaller than a quarter of the catchment area, increased laterally upslope from the stream, and increased to a maximum of two thirds of the catchment area. The simulated connected area during a relatively small event (event I, total rainfall 17 mm) increased by a fifth of the catchment area, which implies that even small rainfall events can activate a sizable part of the catchment. The connectivity simulations for event II, however, suggest that during long duration low-intensity rainfall events, the change in connectivity can be small. For this event, the relative contributions of soil water and rainfall to stormflow were much higher than for the other events (Table 4).

Using a combination of different tracers to identify the sources of streamflow can be helpful, because it enhances the likelihood that sources that contribute little to stormflow are identified (Barthold et al., 2017), and thereby reduces the risk of false conclusions about catchment functioning (Barthold et al., 2011). For instance, McCallum et al. (2012) used differential flow gauging and conservative (Cl) and non-conservative (Rn and EC) tracers to quantify the inflows and outflows of groundwater along three ~30 km long stream reaches in the Cockburn River, Logan River and Nambucca River catchments ($> 400$ km$^2$) in southeast Australia. They found that predictions made with flow data alone varied significantly from predictions that also included tracer data, and that the use of multiple tracers reduced the error in the calculation of the groundwater contributions. Moreover, the discrepancy between the results of source-area analyses based on conservative and non-conservative tracers are hypothesized to indicate when other sources than baseflow and rainfall contribute to streamwater (Kirchner, 2003). We found that the event-water fractions from two-component hydrograph separation (isotopes) and EMMA (multi-tracer) were comparable (Table 4). Similar to our results, Ladouche et al. (2001) found for the 0.8 km$^2$ Strengbach catchment in France that the hydrograph separation results based on $\delta^{18}$O ($f_{pe}$: 10%) were relatively similar to the results of their mixing analyses (including DOC, Si, Ba, and U), and that a multi-tracer approach allowed them to distinguish between pre-event water contributions from the upper and lower part of the catchment. We found that concentrations of metals, such as iron or copper, were much higher than expected from mixing of rainfall and baseflow, whereas weathering-derived solutes, such as sodium or calcium, were lower than expected from mixing of rainfall and baseflow. We assume that the differences between measured and expected concentrations, particularly on the falling limb and at peak flow, are at least partly caused by contributions from groundwater sources or soil water (particularly for event II) that did not contribute to baseflow (see Table 3 for ratios of concentrations in different source waters). For instance, the differences for weathering-derived solutes could be due to contributions from soil water, which has lower concentrations of these solutes than groundwater. The concentrations of iron increased throughout the event until peak flow and were higher on the falling limb than on the rising limb. Since riparian groundwater has relatively high concentrations of iron (Table 2 and 3), contributions from riparian-like areas that did not contribute to baseflow (such as flatter areas away from the stream network) during rainfall events could explain this increase. Measured copper concentrations were much higher than expected for events III and IV, but lower than expected for most samples of events I and II. Because copper concentrations are relatively high for hillslope groundwater and low in soil water (Table 2 and 3; Kiewiet et al., 2019), this could be an indication that the hillslopes did not actively contribute to streamflow during event I and II, and were only activated after peak flow for events III and IV (see wide hysteresis for event I in Fig. 5, top row). However, the copper concentrations should then also not have increased compared to baseflow during event II, which was not the case (maximum $R_{Cu}$ during event II: 1.7 vs 1.0, 1.0 and 1.4 during event I, III and IV, respectively). The potassium

concentrations were too variable to aid further interpretation, which is probably due to the high variation in potassium concentrations in soil water and groundwater (Table 2).

The contribution from soil water was considerable ($f_{SW}$: 0.27) for only one of the four events (event II, Table 4). This was a long, low intensity event, occurring on a relatively 'dry' catchment (baseflow event I and II: 0.2 mm h$^{-1}$ vs. 0.7 mm h$^{-1}$ for event III and IV). Hagedorn et al. (2000), analysed three rainfall events (7, 8 and 30 mm) in the neighbouring Erlenbach catchment and showed a large contribution of soil water to streamflow. Their mixing diagrams using chloride and calcium indicate that the average contribution of the top soil to streamflow was larger than 50%. However, chloride and calcium concentrations vary considerably in both soil and groundwater (average coefficient of variation: 0.86 and 1.0 for eight soil water (n=6 to 18) and 1.0 and 0.3 for nine groundwater (n=34 to 47) snapshot campaigns for chloride and calcium respectively). Furthermore, the concentration of bivalent cations, like calcium, in rainwater can increase during transport through the canopy (Lindberg et al., 1986). van Meerveld et al. (2018) showed that calcium concentrations in overland flow from small landslide areas in the Studibach were much higher than for other solutes, indicating rapid dissolution as well. The much lower soil water contributions found for this study compared to Hagedorn et al. (2000) may thus be partly caused by the choice of the tracers. Understanding the role of soil water for runoff generation processes is challenging because of the spatial variation in its amount (e.g., McMillan and Srinivasan, 2015), the horizontal and vertical spatial variation in soil water chemistry (Gotteselig et al., 2016), and the importance of preferential flow (e.g., Wiekenkamp et al., 2015). Antecedent soil moisture conditions also affect runoff amounts and stream chemistry (Zehe et al., 2010; Uber et al., 2018; Knapp et al., 2020), as well as hillslope-stream connectivity (Penna et al., 2011). Further investigation of the response of soil water, the distribution of soil water chemistry and the interaction between soil water and groundwater during rainfall events is thus important if we want to understand the influence of soil water on hydrologic connectivity and when and where soil water contributes to streamflow.

The typically moderate event-water fractions could indicate that overland flow is of minor importance for streamflow in the Studibach. However, overland flow does occur in the Studibach (van Meerveld et al., 2018). Saturation overland flow has been observed during sprinkling events for other sites on gleysols in Switzerland as well (Feyen et al., 1996; Weiler et al., 1999; Badoux et al., 2006). Given the low event-water fractions, we suspect that the overland flow mixes with pre-event soil water on its way to the stream (Kienzler and Naef, 2008; Elsenbeer and Vertessy, 2000), or originates from exfiltrating soil water or groundwater and thus does not have the same composition as rainwater (Barthold et al., 2017). Alternatively, overland flow may infiltrate in unsaturated soils before reaching the stream, and thus not influence the streamwater composition.

**5.3 Hydrologic connectivity and streamwater chemistry**

The simulations of the active and connected area suggest that the near-stream areas are most often connected and respond first to rainfall, highlighting their importance for the rapid generation of streamflow. The model results also showed that some areas remain disconnected from the stream (Fig. 4). Nippgen et al. (2015) found very similar connectivity patterns for a subcatchment of the Tenderfoot Creek Experimental Forest (5.55 km$^2$) in central Montana, USA. They simulated the connected area over a two-year period and found that it expanded from areas parallel and close to the stream during low-flow conditions, to the hillslopes during high-flow conditions, and that 10% of the catchment was never connected to the stream.

The change in streamwater chemistry also suggests that the connected area increased rapidly because even for small increases in discharge, stormflow could not be described as a mixture of rainfall and baseflow. However, there was no clear relation between the extent of the hydrologically connected area and the discrepancy between the relative changes in the concentrations of conservative and non-conservative solutes (Fig. S5). Other studies that used streamwater chemistry to investigate hydrological connectivity focused on one tracer that was clearly different for different source areas (e.g., Soulsby et al., 2007;

Ocampo et al., 2006). These studies illustrated that for some catchments the changes in streamwater chemistry reflect changes in hydrological connectivity. However, other studies showed that the interpretation of stream-based measurements may not always be straightforward because the changes in streamwater chemistry can be obscured by dampening and mixing processes (Tezlaff et al., 2014), or because a tracer might only reflect connectivity to a specific part of the catchment, rather catchment-wide connectivity (e.g., areas with high DOC concentrations for Pacific et al., 2010). For instance, Pacific et al. (2010) compared changes in streamwater DOC concentrations with estimates of upslope-riparian-stream (URS) connectivity (methods cf. Jencso et al., 2009) in the Tenderfoot Creek catchment. They found a negative (though insignificant) relation between stream DOC export and URS-connectivity, and showed that URS-connectivity is particularly important to predict DOC export when areas with high DOC concentrations are connected to the stream. Multiple studies in the Girnock catchment in Scotland used streamwater Gran alkalinity and isotopic composition to investigate hydrologic connectivity (Soulsby et al., 2007; Tezlaff et al., 2014). Birkel et al., (2010), furthermore, explored the catchment's functioning with a spatially and temporally dynamic saturation model. These studies found that contributions from the upper soil layers and upslope areas dominated streamflow at higher flows and that there was a soil moisture threshold for the contribution of these sources (Birkel et al., 2010). Furthermore, Tezlaff et al. (2014) showed that the dynamic behaviour of the isotopic composition of streamwater was in the range of the composition of soil water from the riparian peat soils at 10 and 30 cm deep , and only deviated from this range during some larger events. They concluded from these results that precipitation inputs drive the dynamics of streamflow and streamwater isotopic composition but that the streamflow responses are dampened because the water travels through different hydropedological units.

Despite substantial changes in the hydrologically connected area and the large spatial variability in groundwater composition, we did not observe a distinct threshold in the relation between the deviation of stream chemistry from conservative mixing of rainfall and baseflow and either streamflow or the connected area. The gradual change in streamwater chemistry might reflect the gradual increase in the connected area with increasing discharge for all of the studied events, except event I, for which the connectivity increased abruptly after peak discharge (top row in Fig. 5). Abbott et al. (2018) showed that changes in streamwater composition with increasing discharge and connectivity are less pronounced for catchments with a myriad of source areas than for catchments with fewer different landscape elements. The Studibach is characterized by many small landscape elements, particularly steep hillslopes and flatter wet areas, which formed due to landslides and soil creep, and which induce small-scale differences in drainage and thus soil and vegetation development. Hence, activation of different landscape elements might occur gradually and at many different places across the catchment (i.e., the connected area extends from flat locations to the hillslopes at many different locations), but these elements all have a slightly different chemical composition. From this perspective, it is perhaps not surprising that solute concentrations in stormflow changed little compared to the spatial variability in the end-member composition because streamflow is a mixture of the many different water sources in a catchment.

Alternatively, the simulations of the active and connected areas might overestimate the change in the source areas compared to reality. Although most flow occurs in the upper, more permeable layer of the soil, seepage to deeper soil layers (Feyen et al., 1999), or to the bedrock in areas where there is no continuous groundwater table in the gleysol, may have decreased the downslope travel distance (cf. Jackson et al., 2014). We did not consider a limitation of the downslope travel distance due to bedrock infiltration because the occurrence of a permanent water table in a large part of the catchment implies that percolation to the bedrock is very slow. However, bedrock infiltration might occur at some locations (e.g., the more densely rooted forested sections on steeper better-drained soils), and might decrease the lateral distance that a water parcel can travel. Additionally, we did not consider an offset in the timing of the simulated connectivity and response in streamwater chemistry due to the travel time to the stream or mixing of hillslope and riparian groundwater in the riparian zone. Chanat and Hornberger (2003) showed with a virtual experiment for a 10 km$^2$ hypothetical catchment that the change in the chemical signature of the

streamwater can be delayed relative to the change in discharge, and that this delay was larger when the near-stream reservoir
(i.e., riparian zone) was larger. Their findings are thus especially important to consider for 'wet' catchments that have a large
near-stream reservoir, or for which the near-stream reservoir expands quickly. Furthermore, the stormflow composition is the
result of mixing of contributions from different source areas. Subsurface mixing can result in temporally variable end-member
compositions. Frameworks to handle time-variable end-member compositions exist (Harris et al., 1995), but there are obvious
challenges, such as measuring these time variable compositions. Furthermore, mixing of different water sources will dampen
the tracer signal (Abbott et al., 2018; Tezlaff et al., 2014) or may even chemically 'reset' the hillslope signal as it mixes with
riparian groundwater (Tezlaff et al., 2014; Lidman et al., 2017).

## 6. Conclusions

The results of this study show that the spatial variability in soil water and groundwater composition across the small pre-alpine
headwater study catchment was large. Hydrograph separation and EMMA indicated that pre-event groundwater was the
545 dominant source of streamflow, and that soil water contributions were minimal for three of the four events. For most solutes,
the streamwater concentrations could not be explained by conservative mixing of baseflow and rainfall. The differences were
largest at high discharge. This suggests that this deviation may indicate the contribution from new contributing sources due to
the expansion of the connected area. Concentrations of weathering-derived solutes decreased more than expected, which might
be due to the contributions of soil water. In contrast, concentrations of iron and copper increased more than expected, which
might be due to contributions from riparian-like areas and hillslopes, respectively. Thus, the differences between the expected
and measured concentrations could be partly explained by contributions from other source areas. However, there was no
threshold in the relation between streamflow and the deviations of the measured concentrations and expected concentrations
based on conservative mixing, suggesting that there was no sudden activation of source areas. The lack of a threshold-relation
between the deviations in the solute concentrations and streamflow made it more difficult to infer changes in hydrological
connectivity from the streamwater solute concentrations. Overall this work shows that inferring hydrological connectivity from
solute concentrations is not straightforward, especially if we consider the large variability of the tracer concentrations in the
different water sources. The gradual changes in streamwater chemistry during events are likely the result of increases in the
contributions from many (small) landscape elements in the catchment and reflect the gradual increase in hydrologic
connectivity.

**Data availability**

The data that support the findings of this study are available from the corresponding author upon reasonable request.

**Author contributions**

LK and IVM conceptualized the study. LK collected and analysed the data, and prepared the first draft of the manuscript. IVM,
JS and MS provided recommendations for the data analysis, participated in discussions about the results, and edited and
565 commented on the manuscript.

**Competing interests**

The authors declare that they have no competing interests.

**Acknowledgements**

This work would not have been possible without the help and support of many people in the field and lab. We particularly thank Michael Rinderer and Benjamin Fischer for the initial installation of the wells, weirs and flumes, Michael Rinderer for running the data-based connectivity model for the Studibach; Barbara Herbstritt for the isotope analyses, and Bjorn Studer for the cation and anion analyses. We thank the editor and two anonymous reviewers for their helpful comments to improve the manuscript, and the Oberallmeindkorporation Schwyz (OAK), the municipality of Alpthal, and the Department of Environment of the Canton of Schwyz for the excellent cooperation.

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

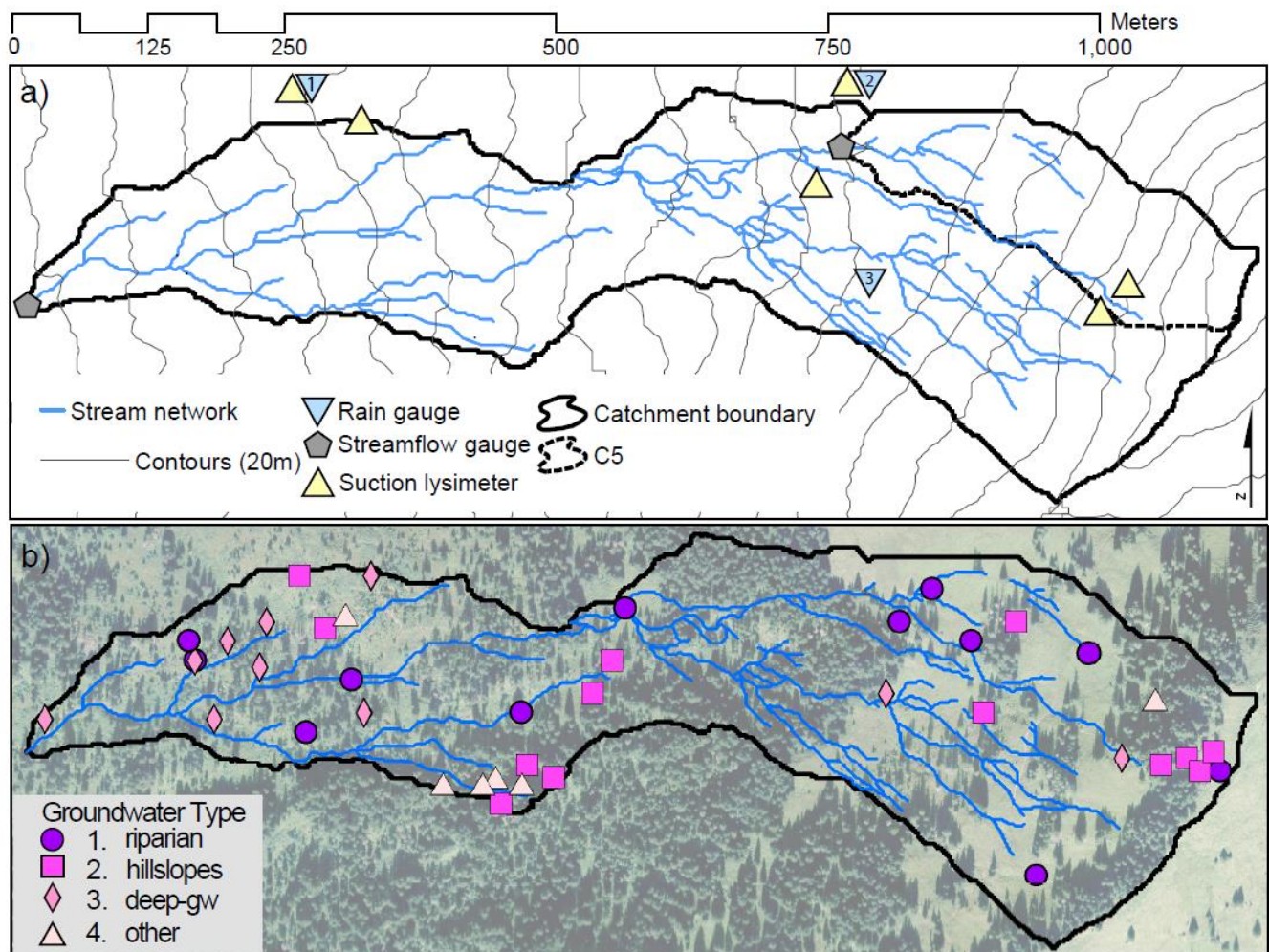

**Figure 1. Maps of the Studibach catchment with a) the stream network (blue lines), stream gauges (grey pentagons), rain gauges (blue reversed triangles, 1 – 3) and suction lysimeters (yellow triangles), 20 m contour lines (grey) and the boundary of the catchment (black) and C5 sub-catchment (dashed lines) and b) location of the wells, colour coded by groundwater type 1. riparian wells; 2. hillslope wells; 3. 'deep' groundwater wells; 4. wells with high magnesium and sulfate concentrations (based on Kiewiet et al., 2019).**


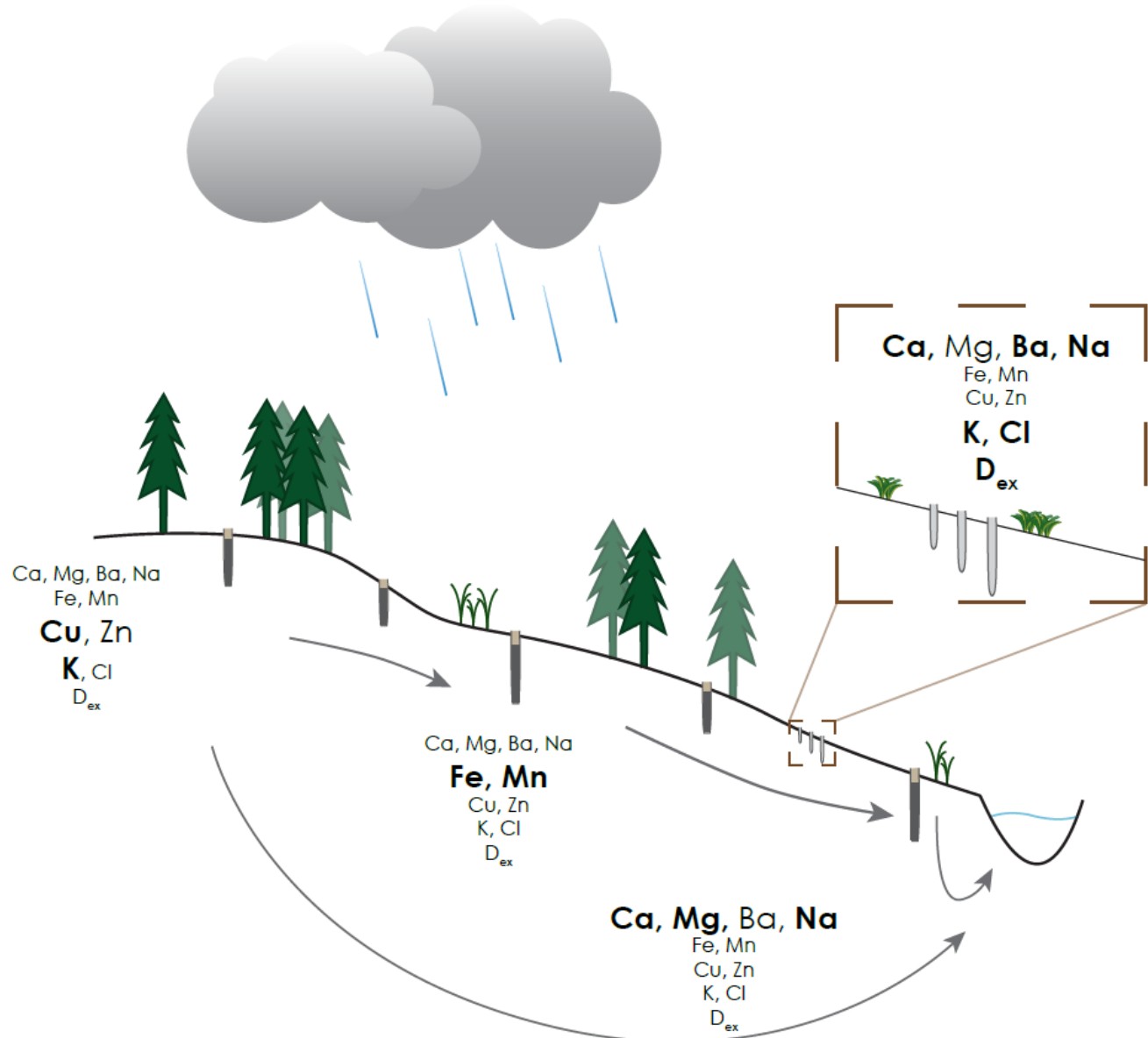

**Figure 2. Illustration of a hillslope cross-section with different (ground)water compartments (based on Kiewiet et al., 2019 and Table 2), showing the tracers used in combination with $\delta^2H$ and $\delta^{18}O$ to characterize the different source areas. For most elements, the concentrations were low in rainfall compared to the concentrations in the other water compartments. High potassium, barium and chloride concentrations and high deuterium excess ($D_{ex}$) are indicative of soil water. For shallow groundwater, the concentrations of copper and potassium were higher at (forested) ridge locations, whereas for sites with water tables that are persistently close to the surface, the concentrations of iron and manganese were higher. We assume that higher concentrations of geogenic solutes (calcium, magnesium and sodium) indicate longer subsurface residence times. The isotopic composition for the different water compartments depends on the composition of recent precipitation.**

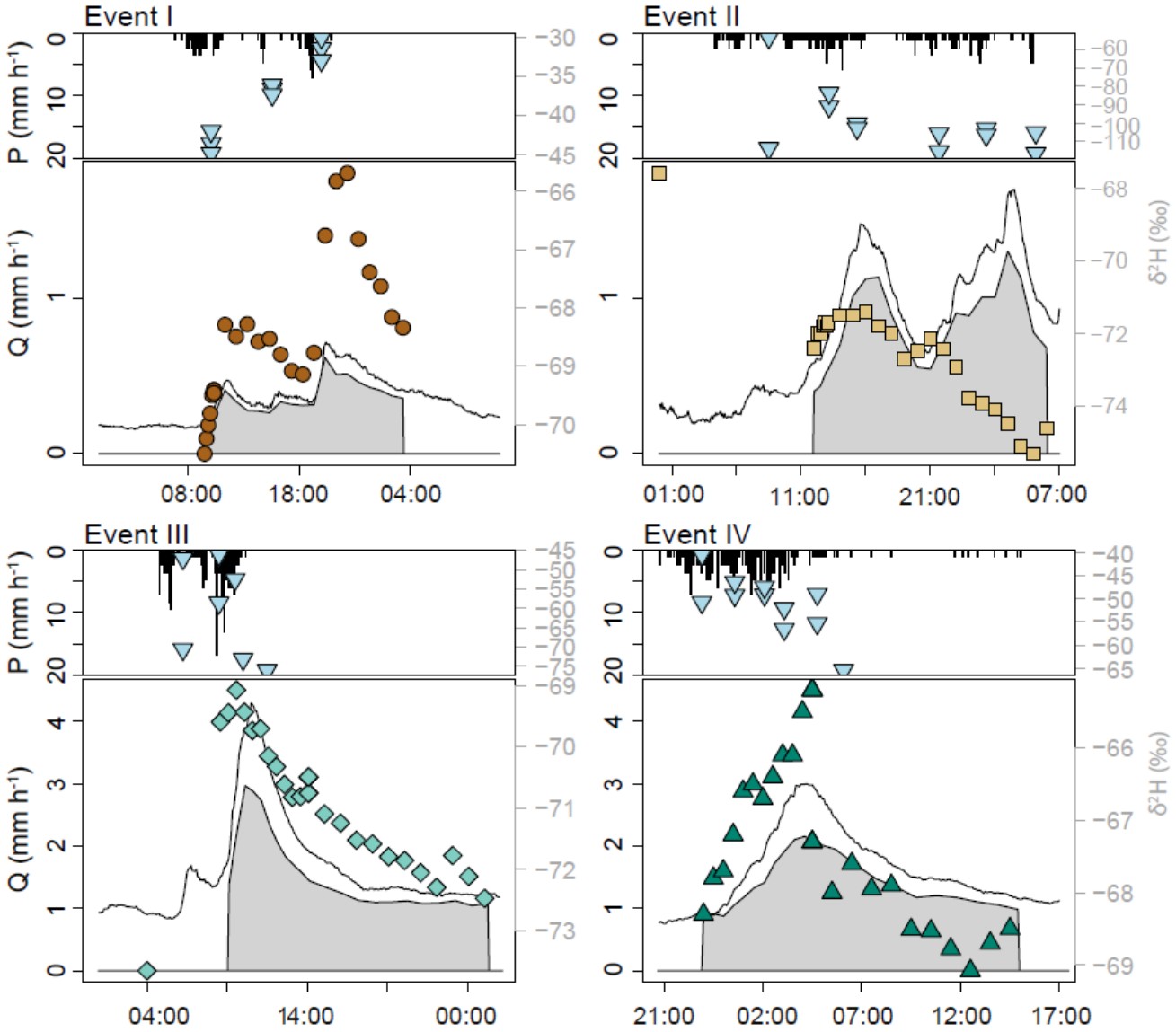

**Figure 3. Hydrographs and hyetographs for the four studied events (I – IV). For each event, the upper panel shows the 10-min rainfall intensity (mm h⁻¹, bar graph) and the isotopic composition of the rainfall (δ²H in ‰, light blue reversed triangles), while the lower panel shows the discharge at the catchment outlet (mm h⁻¹, solid line), the isotopic composition of streamwater (δ²H in ‰, brown dots, light brown squares, turquoise diamonds and green triangles for event I-IV, respectively), and the pre-event water fraction of streamflow based on two-component hydrograph separation using δ²H (grey polygon) as a tracer.**


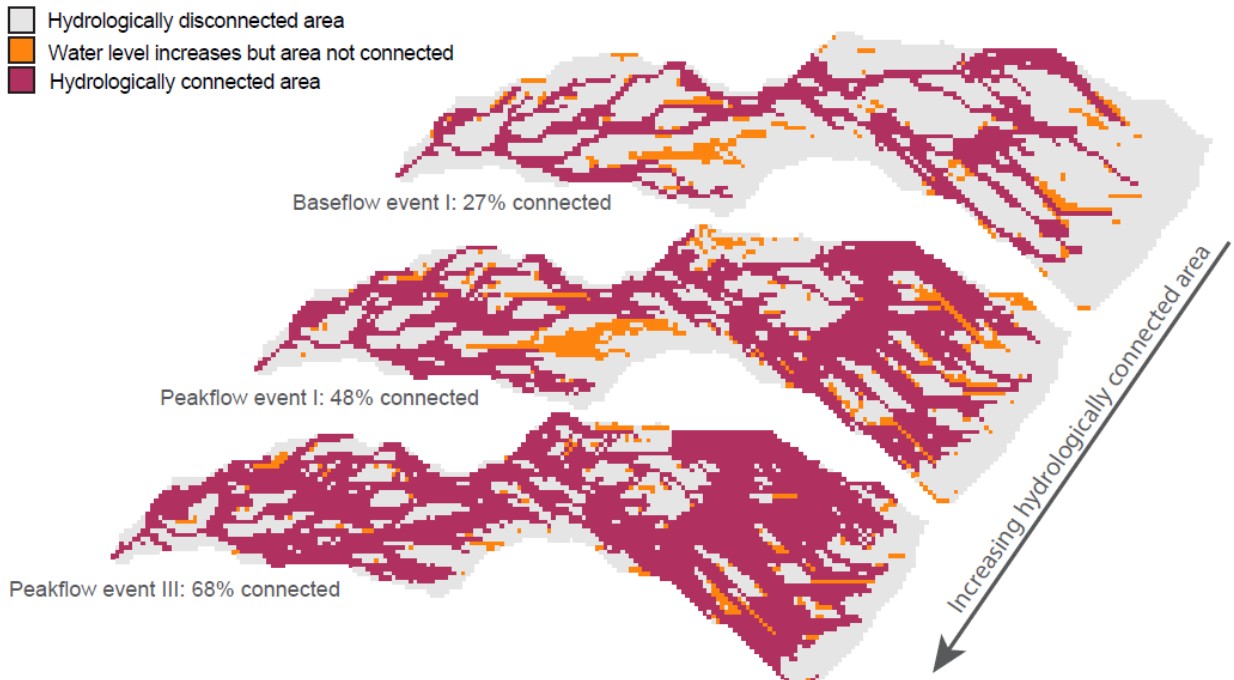

Legend:
- Hydrologically disconnected area
- Water level increases but area not connected
- Hydrologically connected area

Baseflow event I: 27% connected

Peakflow event I: 48% connected

Peakflow event III: 68% connected

Increasing hydrologically connected area

**Figure 4. The simulated hydrologically connected area for three different flow conditions: from relatively low flow (baseflow prior to event I; top), to intermediate flow conditions (peak flow during event I; middle), to the period of highest discharge for the studied events (peak flow during event III; bottom). Grey indicates the hydrologically disconnected areas (water level more than 30 cm from the soil surface), red indicates the hydrologically connected area (i.e., water level within 30 cm from the soil surface and connected to the stream via other active areas), and orange indicates the active but disconnected area (i.e., the water level increased into the upper 30 cm of the soil but is not connected to the stream network by other active areas). The connected area was simulated based on the measured groundwater levels and a data-driven model that uses surface topography to estimate the water level for unmonitored grid cells (cf. Rinderer et al., 2019).**

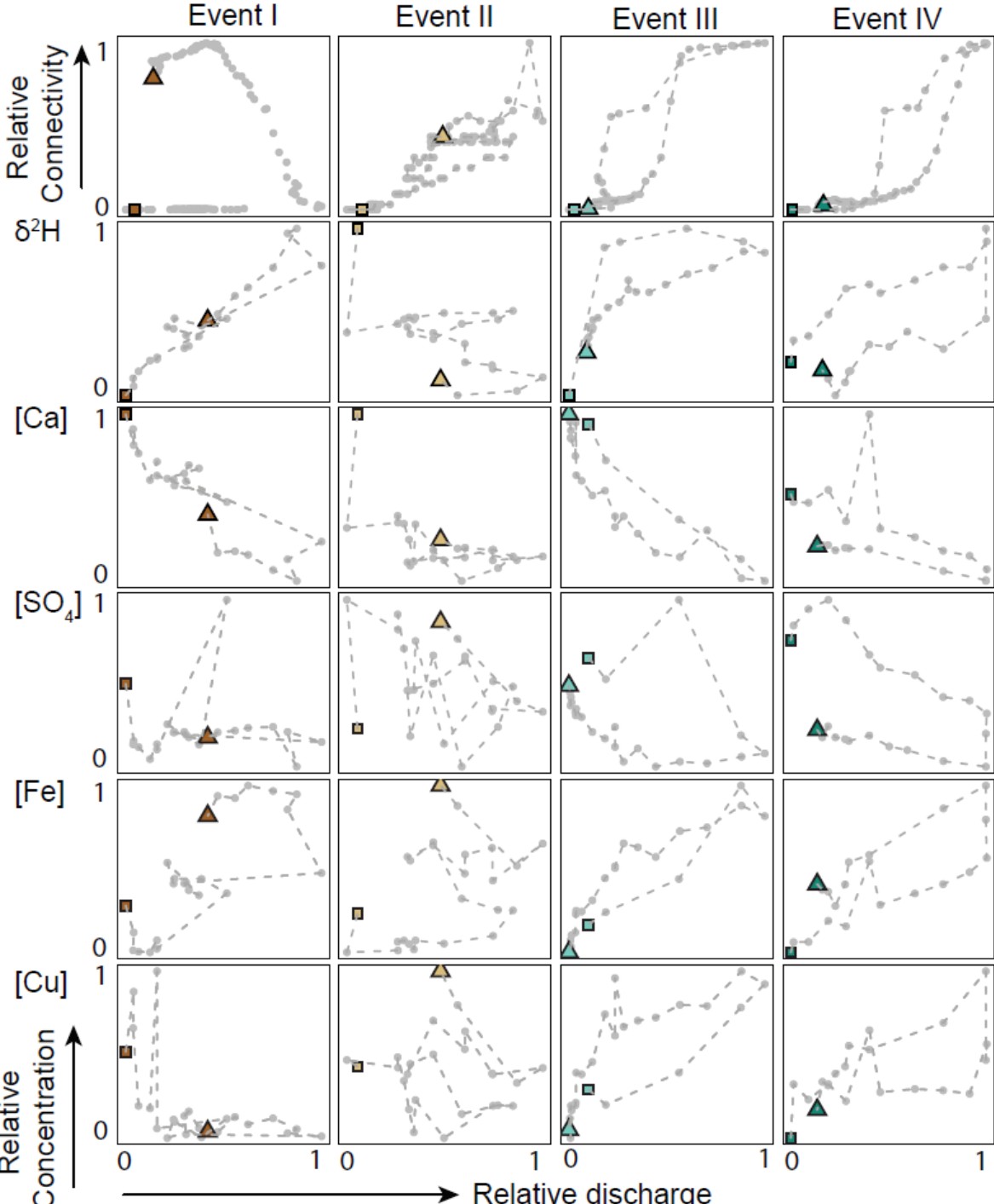

**Figure 5. Relationship between the fraction of the catchment that was connected (relative connectivity) and discharge (top row), and concentration-discharge relationships for δ²H, calcium, sulfate, iron and copper (rows 2-6) for events I-IV (columns). Individual samples are marked with a grey dot and connected with a dashed line, the first sample of the event is indicated by a square, and the last sample by a triangle. All data are normalized between 0 (minimum measured value for the event) and 1 (maximum measured value for the event) for better visualization of the hysteretic relation.**

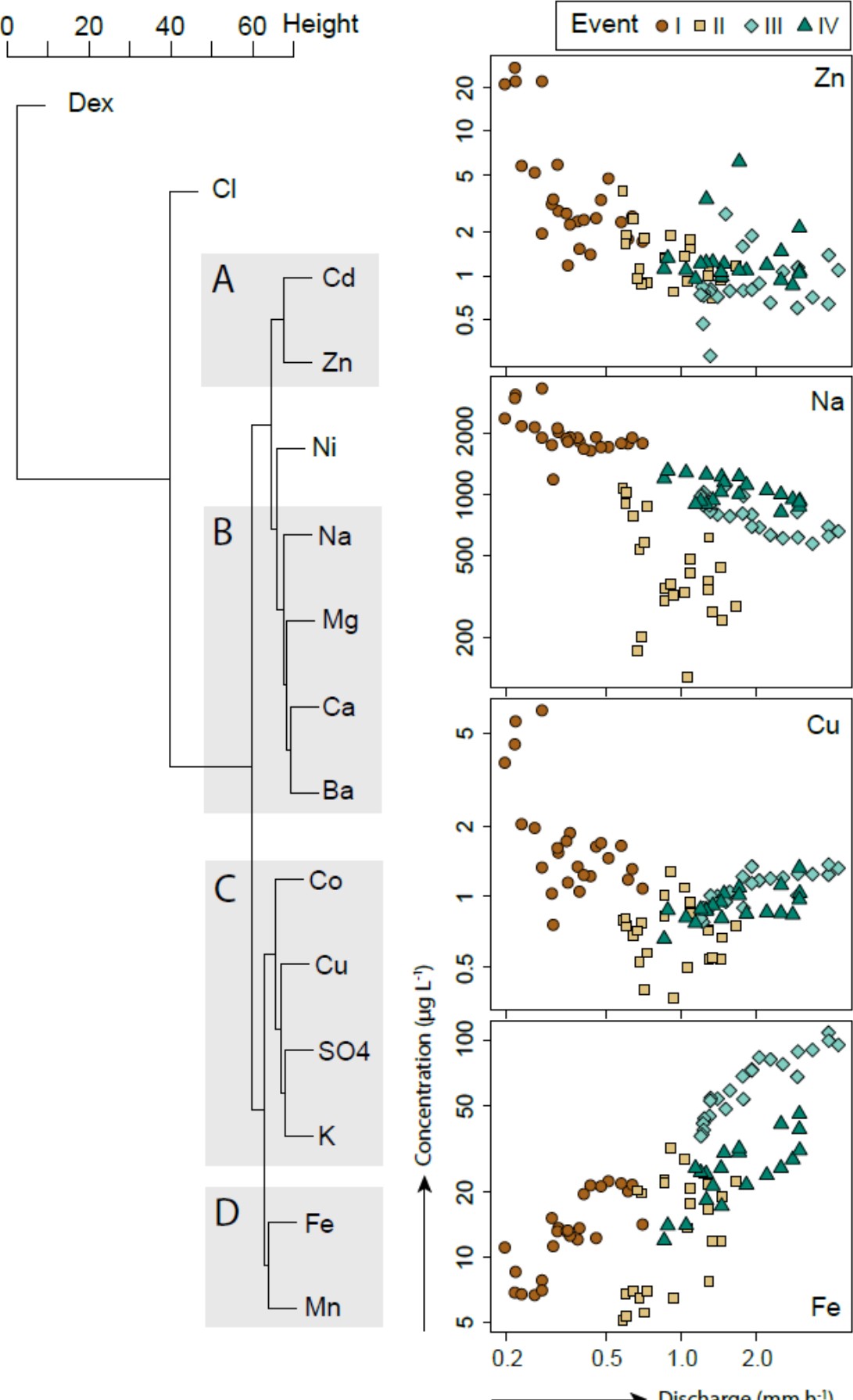

Figure 6. Dendrogram for the hierarchical clustering of solutes and $D_{ex}$ based on the magnitude and timing of changes in streamflow concentrations compared to the baseflow concentration ($R_x$; Eq. 2) during the four events (I-IV), and concentration-discharge relationships for one solute from each group (A-D).

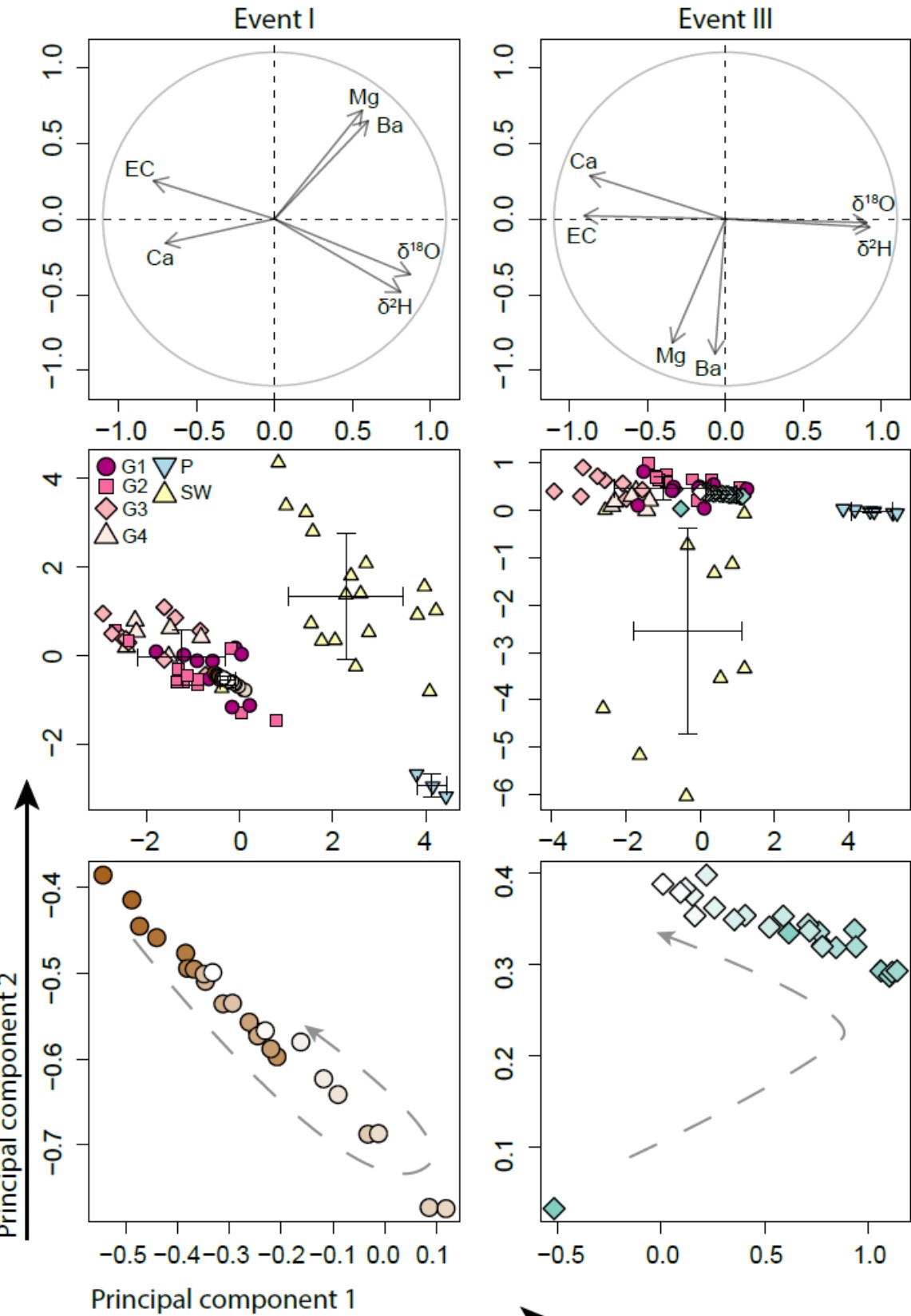

**Figure 7. PCA results and mixing diagrams for events I (small event) and III (intermediately sized event). In the biplots (top row), the length of the arrow represents the explanatory power. The mixing diagrams based on the first two principal components (middle row) show the individual rainfall (light blue reversed triangles), soil water (yellow triangles), and groundwater samples (purple circles, pink squares, light pink diamonds and rose triangles, representing groundwater types 1-4 based on Kiewiet et al., 2019), the streamflow (Q) samples, as well as the average and standard deviation for each component (error bars). The third row shows a zoom-in of the streamflow samples and highlights the evolution of the streamwater composition (colours fade to white towards the end of the event); the general direction of change is indicated with a grey arrow and dashed lines. The biplots and mixing plots for the events II and IV are shown in supplementary material S4.**

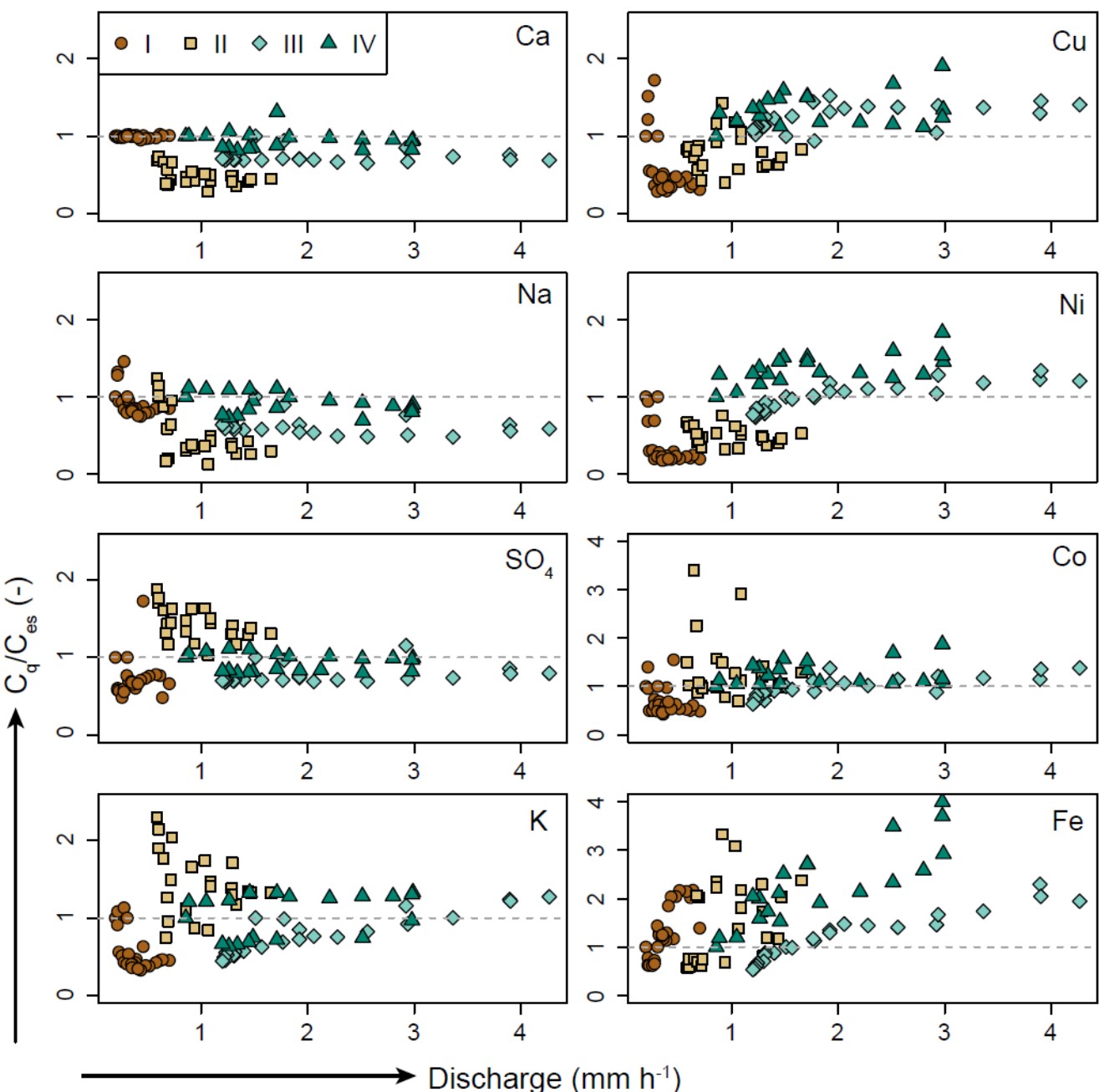

Figure 8. The ratio of the measured ($C_Q$) and estimated stormflow concentrations ($C_{es}$; Eq. 3) for calcium, sodium, sulfate, potassium, cobalt, copper, nickel and iron as a function of discharge at the catchment outlet. The dashed grey line indicates where $C_Q$ and $C_{es}$ are equal; the different symbols reflect the different events (I-IV). Note the difference in scale for cobalt and iron. For the relation with the simulated fraction of the catchment that was connected to the stream see Figure S5.

**Table 1. Overview of the four events analysed in this study: event duration (D, h), rainfall amount (P, mm), average and maximum 10-min rainfall intensity ($I_p$ and $I_{p-max}$, mm h$^{-1}$), the maximum change in specific discharge ($\Delta Q$, mm h$^{-1}$), the maximum change in isotopic composition of the streamwater ($\delta^2H$, ‰), and the minimum and maximum fraction of the catchment that was connected ($A_{min}$-$A_{max}$) during the event, and the date of the groundwater and soil water sampling campaign.**

| Event | Start date | $D$ [h] | $P$ [mm] | $I_p$ [mm h$^{-1}$] | $I_{p-max}$ [mm h$^{-1}$] | $\Delta Q$ [mm h$^{-1}$] | $Q$-$\delta^2H$ [‰] | $A_{min}$-$A_{max}$ [-] | Date of sampling campaign |
|---|---|---|---|---|---|---|---|---|---|
| I | 02 Oct 2016 | 14 | 17 | 1.2 | 7 | 0.02 – 0.07 | -70.5 to -65.7 | 0.27 – 0.48 | 05 Oct. 2016 |
| II | 25 Oct 2016 | 28 | 33 | 1.2 | 13 | 0.02 – 0.17 | -75.3 to -67.6 | 0.27 – 0.35* | 05 Oct. 2016 |
| III | 03 Oct 2017 | 7 | 27 | 3.9 | 24 | 0.08 – 0.43 | -73.7 to -69.1 | 0.33 – 0.68 | 12 Oct. 2017 |
| IV | 05 Oct 2017 | 27 | 32 | 1.2 | 10 | 0.07 – 0.30 | -69.1 to -65.2 | 0.33 – 0.67 | 12 Oct. 2017 |

*The fraction of the catchment that was hydrologically connected increased from 0.27 to 0.28 during the sampling period, and to 0.35 during a discharge peak that occurred after the samplers stopped (see S3).

**Table 2. Average concentrations (± standard deviation) for all groundwater ($GW_{avg}$; n=335), all riparian groundwater (G1; n=99) and all hillslope groundwater (G2; n=99), soil water (SW; n=116), and rainfall samples (P; n=156). Solutes are ordered by their respective groups (section 4.3; Figure 6). Different superscript letters [a-d] indicate significantly different average concentrations.**

| Solute | Unit | $GW_{avg}$ | G1 | G2 | SW | P |
|---|---|---|---|---|---|---|
| $\delta^{18}O$ | ‰ | -11.0±0.9 [a] | -10.8±1.0 [ab] | -10.9±1.1 [ab] | -10.4±1.6 [a] | -12.3±4.0 [c] |
| $\delta^2H$ | ‰ | -76.0±7.5 [b] | -74.3±8.0 [ab] | -74.9±9.1 [ab] | -70.8±12.4 [a] | -84.4±33.0 [c] |
| $D_{ex}$ | ‰ | 12.0±0.8 [a] | 12.4±0.8 [a] | 11.8±0.9 [a] | 12.0±2.4 [a] | 14.1±3.2 [b] |
| Cl | µg L$^{-1}$ | 830.8±1076.5 [a] | 708.8±570.1 [a] | 890.5±804.9 [ab] | 1070.3±1026.6 [ab] | 327.1±348.7 [c] |
| Zn | µg L$^{-1}$ | 593.9±1745.7 [ab] | 720.4±2218.7 [a] | 698.5±843.8 [ab] | 23.3±12.5 [c] | 19.3±43.0 [c] |
| Cd | µg L$^{-1}$ | 0.05±0.08 [ac] | 0.0±0.1 [a] | 0.1±0.1 [b] | 0.03±0.06 [a] | 0.1±0.2 [bc] |
| Ni | µg L$^{-1}$ | 3.2±4.1 [d] | 1.7±1.4 [ab] | 5.6±6.6 [c] | 2.5±1.5 [ad] | 0.3±0.3 [b] |
| Na | µg L$^{-1}$ | 1587.6±2672.7 [b] | 1107.1±1000.8 [ab] | 827.6±341.3 [ac] | 839.1±565.0 [ac] | 148.7±153.5 [c] |
| Mg | µg L$^{-1}$ | 2235.7±1730.3 [a] | 1292.5±684.3 [ab] | 1164.1±435.6 [ab] | 13612.8±10924 [c] | 26.6±18.9 [b] |
| Ca | µg L$^{-1}$ | 56993.7±21966.1 [b] | 44794.0±17097.6 [a] | 55624.6±18099.0 [b] | 22261.7±27287.8 [c] | 213.4±202.7 [d] |
| Ba | µg L$^{-1}$ | 99.2±171.6 [a] | 64.2±115.2 [a] | 112.3±258.6 [a] | 37350±27637 [b] | 4.8±11.8 [a] |
| Co | µg L$^{-1}$ | 0.8±1.05 [a] | 1.1±1.0 [a] | 0.3±0.2 [bc] | 0.9±1.1 [a] | 0.02±0.02 [c] |
| Cu | µg L$^{-1}$ | 64.9±143.7 [c] | 7.4±16.1 [a] | 175.5±211.8 [b] | 5.2±9.0 [a] | 1.4±1.0 [a] |
| SO$_4$ | µg L$^{-1}$ | 3600.0±5112.5 [b] | 2511.6±2843.2 [ab] | 2418.7±1848.2 [ab] | 1602.0±3061.9 [ac] | 623.1±980.1 [c] |
| K | µg L$^{-1}$ | 530.1±428.0 [bc] | 328.3±219.2 [ab] | 670.3±543.4 [cd] | 754.1±970.8 [c] | 92.2±91.9 [a] |
| Fe | µg L$^{-1}$ | 390.7±1271.1 [ab] | 608.3±1648.4 [a] | 25.4±38.6 [b] | 254.3±775.9 [ab] | 3.5±7.1 [b] |
| Mn | µg L$^{-1}$ | 592.4±1111.6 [c] | 1007.8±911.3 [a] | 68.4±100.5 [b] | 139.9±326.2 [b] | 1.3±1.4 [b] |

**Table 3. Summary of the groups of the solutes (A-D, based on the relative concentrations during all four events; Fig. 6; NG indicates that this solute is not assigned to a group), the typical response of solute concentrations to increasing discharge (++: strong enrichment, mean $R_x > 1.5$; +: enrichment, mean $R_x$ between 1 and 1.5; -: dilution, mean $R_x < 1$; ±: mixed response) and ratios between the average concentrations in soil water ($C_{SW}$) and groundwater ($C_{GWavg}$) and the groundwater from riparian wells ($C_{G1}$) and hillslope wells ($C_{G2}$) (see Table 2). See Fig. 5 and 6 for example concentration - discharge relations for each group of solutes. The solutes are sorted according to their typical response.**

| Solute | Group | Typical [C] response to increasing Q | $C_{SW}/C_{GWavg}$ | $C_{G2}/C_{G1}$ |
|---|---|---|---|---|
| $D_{ex}$ | NG | ++ | 1 | 1 |
| Cl | NG | ++ | 1.3 | 1.3 |
| Fe | D | + | 0.7 | ~0 |
| Mn | D | + | 0.2 | 0.1 |
| Co | C | ± | 1.1 | 0.3 |
| Cu | C | ± | 0.1 | 23.7 |
| $SO_4$ | C | ± | 0.4 | 1 |
| K | C | ± | 1.4 | 2 |
| Cd | A | ± | 0.6 | - |
| Zn | A | ± | ~0 | 1 |
| Ni | NG | ± | 0.8 | 3.3 |
| Na | B | - | 0.5 | 0.7 |
| Mg | B | - | 6.1 | 0.9 |
| Ca | B | - | 0.4 | 1.2 |
| Ba | B | - | 376.5 | 1.7 |

**Table 4. Event-average pre-event water fraction ($f_{pe}$) based on the two-component hydrograph separation using $\delta^2H$ as a tracer, and the event average fractions of groundwater ($f_{GW}$), soil water ($f_{SW}$), and rain water ($f_P$), based on the three-component End-Member Mixing Analyses, the, and the associated uncertainties for both calculations.**

| Event | Two-component | | Three-component End-Member Mixing Analyses | | | |
|---|---|---|---|---|---|---|
| | $f_{pe}$ | uncertainty | $f_{GW}$ | $f_{SW}$ | $f_P$ | uncertainty |
| I | 0.86 | 0.28 | 0.81 | ~0 | 0.19 | 0.16 |
| II | 0.76 | 0.61 | 0.49 | 0.27 | 0.24 | 0.14 |
| III | 0.81 | 0.69 | 0.72 | 0.01 | 0.27 | 0.16 |
| IV | 0.78 | 0.25 | 0.74 | 0.01 | 0.25 | 0.14 |