# Peer review of "Do streamwater solute concentrations reflect when connectivity occurs in a small pre-alpine headwater catchment?"

_Hydrology and Earth System Sciences, 2019_

## Referee Comment (RC1) · Anonymous Referee #1 · 6 Feb 2020

This paper investigates spatio-temporal variability of end-member chemistry in a mountainous catchment. In a second step, EMMA is performed for four runoff events determining that soil sources contribute in addition to baseflow and precipitation, but groundwater being the dominating component. Additionally, the authors tested whether concentration of geochemicals could be calculated from conservative mixing. This was not the case. The authors also discussed the potential link between chemistry and changing hydrological connectivity.

I find the study and the data set quite interesting. The paper is well written and data analysis is clearly described. While I like to overall paper good, there are several

limitations. 1. While I like the research questions and the introduction, I do not think that the research gaps for questions 1 and 2 are convincingly presented. For question 3 (first part), I believe that literature shows that this is not the case for most catchment where three component EMMA is performed. 2. The connectivity part is a little bit weak. It is only loosely linked to the results and could be made stronger in results and discussion. The study also lacks a clear definition of hydrological connectivity. Is it mass transport here? As connectivity here is linked to GW level rising close to the surface. Several recent papers challenged such a simplified assumption (e.g. Jackson et al., 2014, Klaus and Jackson, 2018, Gabrielli and McDonnell, 2020). I guess this is still somewhat in the debate, but clearly data on bedrock permeability should be presented to check whether the assumed connectivity from GW levels can be realistic. Maybe other proof can be provided that GW level can be used to infer connectivity? 3. While I think that the paper is quite good, the discussion is currently weak. While the authors are discussing the data and their variation in detail (which is appreciated), I miss discussion of the broader impact of the study, as well as a better link to the introduction or the literature in general. Right now, the discussion refers to only a few studies, mainly related to processes in the same catchments. The authors need to present the broader implication of their work, and make their general contribution to the state of the art outside their study site clearer. At the end of the read I was a little unsure on the take home message. I really think the impact of the paper would be much better if that is achieved. 4. The majority of the figures need to be reworked (3, 5, 6, 8). They lack the quality that is needed for publication.

Minor comments: L35: typo "McGuire" L47: The authors present catchment size and location for the Maimai; one could do the same for the Rietholzbach. The introduction generally good; the research gaps for the first two research question should be made more clear. L92: Why should it only be baseflow? The literature is quite clear that, if tested, this is barely the case. So why asking a question we know to be not true? L150: That is a valid assumption; but how variable is soil water chemistry (yes, the data is partly presented, but it could be stated)? Additional some more information on the

choice of geochemicals and their commonly observed behaviour would be nice. L188: A clear definition of connectivity is needed, especially when not investigating the mass flux directly. L198/199: You can only assume connectivity in cases where one have a low permeable of underlying bedrock (cf. Jackson et al., 2014; Klaus et al., 2018; Gabrielli and McDonnell, 2020). L219: Define "similar" L251ff: There is a nice paper by Harris et al. (1995) that looked into changing end-member contributions. The idea is not too different from the one here. L251ff: There is a range of studies that looked (e.g. McCallum et al., 2010), related to hydrograph separation, how GW chemistry is different from baseflow chemistry. L345: Or does that indicate a much less pronounced connectivity compared to the model? L365: Is that surprising? The spatial variability is the maximum extend of the mixing diagram of endmembers. Thus, changes in the stream must be smaller, if the sampling was representative. Discussion I am missing the bigger picture here. The discussion is very detailed and evolves around the data being non-conclusive. It would be nice to expand this section and discuss what the key contribution to the field of runoff generation is. How do you go beyond studying this catchment? How does your work related to previous work? What is the key novelty? You may also think of linking your discussion better to the introduction and the used references there. L448: but for some? And what do you infer from that? Figures 3, 5, 6, 8 are not very well done. While the content is fine, the presentation, choice of colours, font size, and point type should be revised.

References

Gabrielli, C.P., McDonnell, J.J. (2020): Modifying the Jackson index to quantify the relationship between geology, landscape structure and water transit time in steep wet headwaters. Hydrological Processes, in press.

Harris, D.M., McDonnell, J.J., Rodhe, A. (1995): Hydrograph separation using continuous open system isotope mixing. Water Resources Research 31 (1), 157–171.

Jackson, C.R., Bitew, M., Du, E. (2014): When interflow also percolates: downslope

travel distances and hillslope process zones. Hydrological Processes, 28 (7), 3195-3200.

Klaus, J., Jackson, C.R. (2018): Interflow Is Not Binary: a Continuous Shallow Perched Layer Does Not Imply Continuous Connectivity. Water Resources Research, 54, 3988–4008.

McCallum, J.L., Cook, P.G., Brunner, P., Berhane, D. (2010): Solute dynamics during bank storage flows and implications for chemical base flow separation. Water Resources Research 46 (7), W07541.

---

## Referee Comment (RC2) · Anonymous Referee #2 · 22 Feb 2020

The manuscript entitled "Do streamwater solute concentrations reflect when connectivity occurs in a small pre-alpine headwater catchment?" by Leonie Kiewiet, Ilja van Meerveld, Manfred Stähli and Jan Seibert, presents an important contribution to the understanding of the hydrological connectivity (or non-connectivity) processes that occur in a pre-alpine catchment, monitored at event scale. The authors presented an exploratory analysis of the hydro-chemical composition of potential water sources and streamflow. They applied widely used, though not so novel, methodologies (simple hydrograph separation and EMMA), but complemented the analysis with hydrological connectivity simulations that make this study interesting. The work is well written, clearly structured and personally enjoyed reading it. Despite the short monitoring pe-

riod, I find it with potential for publication in HESS after addressing a few suggestions. • The concept of baseflow depends on the method used to estimate it and does not always describe active groundwater flow pathways. I suggest the authors describe what they defined in this study as baseflow. • The third objective could be modified, it is well known that baseflow and rain mixture (negligible contribution of soil water) does not explain the changes in solutes concentrations in the streamflow. • One of the principles of EMMA is that it relies on conservative tracers (not involved in adsorption or biological processes) and linear mixing process (Hooper, 2001). Did you analyse the conservative behaviour of the tracers? Please include the tests and state what tracers were used. Also, a graph showing the spatial-temporal concentrations of tracers in water sources would help the reader to contextualize their interaction during events. • Regarding EMMA's analysis, I suggest examining the evolution of events in the PCA space (Inamdar et al. (2013); Barthold et al. (2017); Correa et al. (2018)). Their dynamics and hysteresis can show the proximity of the streamflow to a certain source in the different stages of the event. Although as "soft data" it can bring insights into what groundwater or soil water contributes at a certain time. • I am concerned about the very high uncertainties (Table 4), 160% in event III and 143% in event IV. Could it be due to the limited streamflow data, input-data uncertainty or time-dependent end-member variability (Chaves et al., 2008; Christophersen and Hooper, 1992). Unluckily end-member solutions do not exhibit low variability compared to the stream chemistry and not exhibit distinctive concentrations between end-members. I encourage the authors to analyse this limitation in more detail. • As an alternative the authors could refer to: Phillips, D. L. and Gregg, J. W.: Uncertainty in source partitioning using stable isotopes, Oecologia, 127(2), 171–179, doi:10.1007/s004420000578, 2001, to compute individual uncertainties in the calculation of source contributions to streamflow, this methodology considers the number of samples. The author could identify whether the uncertainties remain very high. The introduction, methods and results sections are complete and clear to follow, despite some very long sentences that make a little difficult to follow the ideas. However, I find the manuscript poorly discussed. The authors

support their findings in an extremely local context. The study would benefit from a broader perspective, comparing it with other similar ecosystems and/or with studies of the dynamics of water source contribution streamflow during events for example. I assume the figures will be uploaded in a high-quality prior publication. In S1 please include rain and streamflow samples to visualize their distribution (potential streamflow at different colour scale for low, medium and high flows) and check the paper for a few typos.

---

## Author Comment (AC1) · 27 Mar 2020

Reviewer #1

This paper investigates spatio-temporal variability of end-member chemistry in a mountainous catchment. In a second step, EMMA is performed for four runoff events determining that soil sources contribute in addition to baseflow and precipitation, but groundwater being the dominating component. Additionally, the authors tested whether concentration of geochemicals could be calculated from conservative mixing. This was not the case. The authors also discussed the potential link between chemistry and changing hydrological connectivity. I find the study and the data set quite interesting. The paper is well written and data analysis is clearly described. While I like to overall paper good, there are several limitations.

We appreciate the overall positive assessment of our work and the helpful suggestions to improve the manuscript.

1. While I like the research questions and the introduction, I do not think that the research gaps for questions 1 and 2 are convincingly presented. For question 3 (first part), I believe that literature shows that this is not the case for most catchment where three component EMMA is performed.

We thank the reviewer for this comment. We recognize that the content and structure of the introduction did not logically lead to the research questions. We will restructure the introduction to provide more background for research questions one and two. We will also add other literature to present the research gaps more clearly.

Regarding the third research question, we agree that the answer to the first part of the question is obvious in some situations. This is exactly why we often use three-component EMMA. We included the first part of the question for completeness but we also see that we need to rewrite the question to emphasize the novelty of our work, rather than reminding the reader about the results from other studies. Therefore, we propose to change the third research question as follows:

"In how far does conservative mixing of baseflow and rainfall explain the changes in stream solute concentrations and do discrepancies from this mixing indicate when other sources become connected to the stream?"

2. The connectivity part is a little bit weak. It is only loosely linked to the results and could be made stronger in results and discussion. The study also lacks a clear definition of hydrological connectivity. Is it mass transport here?

We agree that the discussion of the connectivity part can be improved. We plan to rewrite the discussion (see below) and will include more linkages to the connectivity part. We plan to include a paragraph about the assumption that a connected water table indicates hydrologic connectivity (see below) and discuss the implications of this study for connectivity studies. To improve the link with the results, we plan to expand the paragraph in the discussion that discusses the connectivity results (L412 – 422) as well.

We agree that defining hydrological connectivity is important and that this will help the reader understand the study better. In this manuscript, we refer to connectivity as the flow of water between different locations in the catchment. We plan to include the following definition: "Hydrologic connectivity is the linkage of separate regions of a catchment via water flow" (Blume and van Meerveld, 2015).

As connectivity here is linked to GW level rising close to the surface. Several recent papers challenged such a simplified assumption (e.g. Jackson et al., 2014, Klaus and Jackson, 2018, Gabrielli and McDonnell, 2020). I guess this is still somewhat in the debate, but clearly data on bedrock permeability should be presented to check whether the assumed connectivity from GW levels can be realistic. Maybe other proof can be provided that GW level can be used to infer connectivity?

This is a very interesting question. Klaus and Jackson (2018) indeed showed based on the contrast in soil and bedrock permeability and Darcian flow principles that groundwater will infiltrate the bedrock before reaching the stream in many catchments. However, they focused on situations where there is a perched water table that occurs above the soil-bedrock interface during events. In the Studibach, there is an almost permanent water table in the low conductivity gleysols in most locations. We assume that significant lateral flow occurs when this water table rises into the near-surface layers, where the hydraulic conductivity is much larger (c.f. Schneider et al., 2014). However, flow to deeper soil layers is also substantial (Feyen et al., 1999). We will make it clearer in the manuscript that we talk about groundwater flow in the more permeable layer of the soil, i.e., above a saturated soil.

Gabrielli and McDonnell (2020) calculated which regions of several catchments can or cannot contribute to streamflow for four geologic settings. Both the Klaus and Jackson (2018) and Gabrielli and McDonnell (2020) paper show that the difference between conductivity of the conducting and impeding layer (in these papers soil and bedrock) determines the downslope travel distance, and thus connectivity to the stream. They assume Darcian flow. Preferential flow will significantly increase the upper layer conductivity, and thus increase the distance that a water parcel can travel before infiltrating to the bedrock. Chloride and bromide tracer studies have shown that preferential flow is an important transport mechanism in the Alptal (Feyen et al., 1999). Feyen et al. (1999) estimated that the effective saturated hydraulic conductivity ($K_{sat}$) is roughly $10^3$ times larger than the $K_{sat}$ of the soil matrix (0.0062 m/s vs. 7.2 $10^{-7}$ m/s for the soil at 3-25 cm depth and 0.2 $10^{-7}$ m/s for the soil at >40 cm depth). Therefore, water might be able to laterally transfer much quicker than expected. van Meerveld et al. (2018) report a surface $K_{sat}$ of 2.8-5.6 $10^{-7}$ m/s for the wetland sites (n=2), 5.6 $10^{-7}$-1.1 $10^{-5}$ m/s for a steep meadow site (n=2) and >1.1 $10^{-4}$ in the forest (n=1) (data from Sauter, 2017), which shows that the surface infiltration rate depends on land cover.

If we would assume that the low conductivity part of the lower part of the soil profile is unsaturated (which is not the case), we can calculate the downslope travel distance according to Klaus and Jackson (2018) and Gabrielli and McDonnell (2020). Slug tests for the Studibach groundwater wells (Zehnder, 2013) suggest that the $K_{sat}$ of the soil at the soil-bedrock interface ranges from 1.71 $10^{-6}$ to 3.62 $10^{-9}$ m/s (median: 2.33 $10^{-7}$ m/s). If we use these permeability values to calculate the downslope travel distance using Eq. 1 from Gabrielli and McDonnell (2020), assume an average well depth of 1.05 m, a depth of the more conductive soil layer of 0.3 m, and a slope of 30 degrees (which is the average slope in the catchment) or 10 degrees (the slope in the areas close to the stream), we obtain the flow distances in the upper part of the soil (Table 1).

Table 1: Downslope travel distances of a water parcel through the upper layer of the soil before percolating into the lower soil layer if this layer was unsaturated (which is not the case). Each column shows the $K_{sat}$-values that were assumed for the upper (first row) and lower (second row) soil layer, and the downslope travel distance (m) for a slope of 10° (the slope in areas close to the stream) and 30° (the average slope in the catchment).

| | Assuming no preferential flow | | | Assuming only preferential flow | | |
|---|---|---|---|---|---|---|
| $K_{sat}$ upper soil layer (m/s) | 2.33E-7 | 1.71E-6 | | 0.0062 | | |
| $K_{sat}$ lower soil layer (m/s) | 3.62E-9 | 2.33E-7 | 3.62E-9 | 1.71E-6 | 2.33E-7 | 3.62E-9 |
| Downslope travel distance 10° | 8 m | 1 m | 55 m | 423 m | 3102 m | 200 km |
| Downslope travel distance 30° | 14 m | 2 m | 100 m | 768 m | 5634 m | 360 km |

This back of the envelope calculation suggests that the downslope travel distance through the upper soil layer is large enough for a water parcel to reach the stream via preferential flow pathways. This is in part because the flowing stream network density in the Studibach is high (8.5 to 23.9 km/km$^2$ during dry and wet conditions, respectively; van Meerveld et al., 2019) and travel distances to the stream are thus relatively small. If we do not consider preferential flow, the water only reaches the stream if the permeability of the deeper soil is as low as the minimum measured $K_{sat}$. But the vertical gradients are likely much smaller than the unit gradient assumed by Klaus and Jackson (2018) and Gabrielli and McDonnell (2020) because the lower layer is saturated and thus the travel distances would be smaller.

We do not have $K_{sat}$ data for the bedrock but assume that this is even lower than for deeper soil layers, as otherwise there would not be a permanent water table above the bedrock in such a large part of the catchment.

3. While I think that the paper is quite good, the discussion is currently weak. While the authors are discussing the data and their variation in detail (which is appreciated), I miss discussion of the broader impact of the study, as well as a better link to the introduction or the literature in general. Right now, the discussion refers to only a few studies, mainly related to processes in the same catchments. The authors need to present the broader implication of their work, and make their general contribution to the state of the art outside their study site clearer. At the end of the read I was a little unsure on the take home message. I really think the impact of the paper would be much better if that is achieved.

Upon re-reading the manuscript, we also recognize the weaknesses highlighted by the reviewer and the lack of a broader discussion and clear take-home messages. To overcome this, we plan to rewrite the discussion so that it also includes:

- A comparison of the results to literature from other study sites, as we did in the introduction. We think that, for instance, linkages to the studies of Ladouche et al. (2001) and Soulsby et al. (2007) would be useful here.

- A section that shows the broader impact of the study. We think that such a section should include how our results fit with current concepts of hydrologic connectivity (e.g., Blume and van Meerveld, 2015). It should also address the assumptions made with calculations of connectivity as mentioned above (e.g., Jackson et al., 2014).

In addition, we will rewrite parts of the existing discussion and emphasize the take-home message. We can achieve this with a section on the broader impact of our study as the final paragraph, and finish with a take-home statement. This could be something along the following lines: "The

combination of hydrometric and hydrochemical data can be useful to identify hydrological connectivity and aid the interpretation catchment-scale runoff generation. However, we have to take the variability of the tracer concentrations in different water sources into account, as they can be large compared to the change in streamwater concentrations. The observed gradual deviations in the concentrations that are expected based on mixing of baseflow and precipitation are likely the result of increases in the contributions from many (small) landscape elements in the catchment and thus reflect the gradual increase in connectivity during events."

4. The majority of the figures need to be reworked (3, 5, 6, 8). They lack the quality that is needed for publication.

We realize that the font size might not have been sufficiently large and that the contrast of the color scheme was not sufficient when the manuscript is printed. Therefore, we will enlarge the font and points, and revise the color scheme so that the figures will be more readable.

All our figures (except Figure 2) are vector images and thus have fulfill the DPI requirements. However, by including them in the .pdf document the quality was reduced. We will make sure to fulfil the required DPI when uploading the original figures.

Minor comments:

L35: typo "McGuire"

Thank you for pointing out this mistake.

L47: The authors present catchment size and location for the Maimai; one could do the same for the Rietholzbach. The introduction generally good; the research gaps for the first two research question should be made more clear.

We will mention the catchment sizes and locations throughout the manuscript. See our comments above regarding the first two research questions.

L92: Why should it only be baseflow? The literature is quite clear that, if tested, this is barely the case. So why asking a question we know to be not true?

See our comments regarding the third research question above.

L150: That is a valid assumption; but how variable is soil water chemistry (yes, the data is partly presented, but it could be stated)? Additional some more information on the choice of geochemicals and their commonly observed behaviour would be nice.

We recognize that more information on the variability of tracer concentrations in different water sources (particularly in soil water) and the choice of tracers and their characteristics can aid the reader in understanding the manuscript. Therefore, we will add a short description of the tracers and their variability in the last paragraph of the methods section. This section currently describes how we interpret the changes in streamwater solute concentrations.

Additionally, we will follow the suggestion of reviewer two to include a figure that shows the spatial-temporal variability of the tracer concentrations in each water source. We are testing different figure types but most likely it will be a figure that has a panel for each tracer and shows a boxplot per source (Figure S1). We prefer to add this figure to the supplementary information but of course will mention it in the text.

[Figure]

Figure S1: Box plots of the tracer concentrations for the different water types: groundwater (G), rainfall (P), streamflow (Q) and soil water (S). Each boxplot contains all streamflow or rainfall samples taken during the four events presented in this study, or all soil water or groundwater samples taken during the snapshot campaigns used in the study. Isotopic tracers are shown in ‰, values for chemical tracers in µg/L. Please note that y-axes differ for each panel, and that the y-axes of the panels on the bottom two rows are logarithmic.

L188: A clear definition of connectivity is needed, especially when not investigating the mass flux directly.

We agree that adding a definition of connectivity will be helpful (see answer to the comment above).

L198/199: You can only assume connectivity in cases where one have a low permeable of underlying bedrock (cf. Jackson et al., 2014; Klaus et al., 2018; Gabrielli and McDonnell, 2020).

We realize that we did not explicitly mention or comment on this assumption and will address this assumption in the revised manuscript. See also our answer to the comment above.

L219: Define "similar"

The difference in the event water fraction for the two-component hydrograph separation using $\delta^2$H or $\delta^{18}$O as a tracer was 0.05. We will include this information in the manuscript to better define the similarity.

L251ff: There is a nice paper by Harris et al. (1995) that looked into changing end-member contributions. The idea is not too different from the one here.

Thank you for highlighting this paper. We were not aware of it. Indeed, it discusses the calculations that match the idea that we present. Thus incorporating it in our discussion is a good suggestion.

L251ff: There is a range of studies that looked (e.g. McCallum et al., 2010), related to hydrograph separation, how GW chemistry is different from baseflow chemistry.

Thank you for recommending this paper. The McCallum paper indeed shows how baseflow chemistry can be different from groundwater contributions during a rainfall event. It is interesting because they showed through their numerical model and field-observations that the spatial variability of groundwater is important for hydrograph separation. We will certainly include the reference to this paper and also investigate other papers that discuss this topic, such as Chanat and Hornberger (2003) and Jones et al. (2016).

L345: Or does that indicate a much less pronounced connectivity compared to the model?

We appreciate this suggestion, and recognize that indeed, a less pronounced connectivity change might also be a valid reason for a smaller change in streamflow composition than expected. We will include this alternative interpretation in the manuscript and add a more general comparison of the model results and observations. Such a statement could be along the following lines: "The limited change in streamwater composition might also be the effect of a discrepancy between the model results and the actual expansion of the hydrologically connected area, for instance, because the change in connectivity is smaller than predicted by the model. Additionally, the transport of a water parcel to the catchment outlet takes time, and the expansion of the hydrologically connected network within the catchment and change in streamflow composition at the catchment outlet might thus not be synchronous."

L365: Is that surprising? The spatial variability is the maximum extend of the mixing diagram of endmembers. Thus, changes in the stream must be smaller, if the sampling was representative. I am missing the bigger picture here. The discussion is very detailed and evolves around the data being non-conclusive. It would be nice to expand this section and discuss what the key contribution to the field of runoff generation is. How do you go beyond studying this catchment? How does your work related to previous work? What is the key novelty? You may also think of linking your discussion better to the introduction and the used references there.

Indeed, this is not surprising but very few studies have characterized the spatial variability for groundwater and soil water. We mention that this is not surprising in the final part of the discussion (L442) but we recognize that these two statements are rather far apart and will adjust the text accordingly.

L448: but for some? And what do you infer from that?

Indeed, the contribution of soil water is important during some of the events, but not all events, which further complicates the analysis of connectivity based on stream chemistry. We will discuss the implications of soil water contributions to streamflow in the description of the hydrologic functioning (section 5.2).

Figures 3, 5, 6, 8 are not very well done. While the content is fine, the presentation, choice of colours, font size, and point type should be revised.

Thank you for pointing this out. We will improve these figures following the suggestions.

**References;**

Blume T, van Meerveld HJI. 2015. From hillslope to stream: methods to investigate subsurface connectivity. Wiley Interdisciplinary Reviews: Water 2 (3): 177–198 DOI: 10.1002/wat2.1071

Chanat, J. G., and Hornberger, G. M. ( 2003), Modeling catchment-scale mixing in the near-stream zone—Implications for chemical and isotopic hydrograph separation, Geophys. Res. Lett., 30, 1091, doi:10.1029/2002GL016265, 2.

Feyen, H. & Wunderli, H. & Wydler, H. & Papritz, A.. (1999). A tracer experiment to study flow paths of water in a forest soil. Journal of Hydrology. 225. 155-167. 10.1016/S0022-1694(99)00159-6.

Fischer, Benjamin & van Meerveld, Ilja & Seibert, Jan. (2017). Spatial variability in the isotopic composition of rainfall in a small headwater catchment and its effect on hydrograph separation. Journal of Hydrology. 547. 10.1016/j.jhydrol.2017.01.045.

Gabrielli, CP, McDonnell, JJ. Modifying the Jackson index to quantify the relationship between geology, landscape structure, and water transit time in steep wet headwaters. Hydrological Processes. 2020; 1– 12. https://doi.org/10.1002/hyp.13700

Inamdar, S., Dhillon, G., Singh, S., Dutta, S., Levia, D., Scott, D., Mitchell, M., Van Stan, J, and McHale, P. ( 2013), Temporal variation in end-member chemistry and its influence on runoff mixing patterns in a forested, Piedmont catchment, Water Resour. Res., 49, 1828– 1844, doi:10.1002/wrcr.20158.

Jackson, C. R., Bitew, M., & Du, E. (2014). When interflow also percolates: Downslope travel distances and hillslope process zones. Hydrological Processes, 28(7), 3195–3200. https://doi.org/10.1002/hyp.10158

Jones, J. P., Sudicky, E. A., Brookfield, A. E., and Park, Y.-J. ( 2006), An assessment of the tracer-based approach to quantifying groundwater contributions to streamflow, Water Resour. Res., 42, W02407, doi:10.1029/2005WR004130.

Klaus, J., & Jackson, C. R. ( 2018). Interflow is not binary: A continuous shallow perched layer does not imply continuous connectivity. Water Resources Research, 54, 5921– 5932. https://doi.org/10.1029/2018WR022920

Ladouche B, Probst A, Viville D, Idir S, Baqué D, Loubet M, Probst J-L, Bariac T. 2001. Hydrograph separation using isotopic, chemical and hydrological approaches (Strengbach catchment, France). Journal of Hydrology 242 (3–4): 255–274 DOI: 10.1016/S0022-1694(00)00391-7

Sauter, T. 2017. Occurrence and chemical composition of overland flow in a pre-alpine catchment, Alptal (CH). Department of Geography. University of Zurich, Zurich, 79 pp.

Soulsby, C., Tetzlaff, D., Van Den Bedem, D., Malcolm, I. A., Bacon, P. J., & Youngson, A. F. (2007). Inferring groundwater influences on surface water in montane catchments from hydrochemical surveys of springs and streamwaters. Journal of Hydrology, 333(2-4), 199-213. https://doi.org/10.1016/J.JHYDROL.2006.08.016

von Freyberg, J., Studer, B., Rinderer, M., & Kirchner, J. W. (2018). Studying catchment storm response using event- and pre-event-water volumes as fractions of precipitation rather than discharge. Hydrology and Earth System Sciences, 22(11), 5847-5865. https://doi.org/10.5194/hess-22-5847-2018

van Meerveld H.J.I., Fischer B.M.C., Rinderer M., Stähli M., Seibert J. 2018. Runoff generation in a pre-alpine catchment: A discussion between a tracer and a shallow groundwater hydrologist. Cuadernos de Investigación Geográfica 44 (2): 429–452 DOI: 10.18172/cig.3349

van Meerveld, H. J. I., Kirchner, J. W., Vis, M. J. P., Assendelft, R. S., and Seibert, J.: Expansion and contraction of the flowing stream network alter hillslope flowpath lengths and the shape of the travel time distribution, Hydrol. Earth Syst. Sci., 23, 4825–4834, https://doi.org/10.5194/hess-23-4825-2019, 2019.

Zehnder, M. 2013. Hydraulic Conductivity Estimation and Analysis using Slug Tests - Relations with Site-Characteristics in a pre-alpine torrent catchment, Alptal (SZ). Department of Geography. University of Zurich, Zurich

---

## Author Comment (AC2) · 27 Mar 2020

Reviewer #2

The manuscript entitled "Do streamwater solute cocentrations reflect when connectivity occurs in a small pre-alpine headwater catchment?" by Leonie Kiewiet, Ilja van Meerveld, Manfred Stähli and Jan Seibert, presents an important contribution to the understanding of the hydrological connectivity (or non-connectivity) processes that occur in a pre-alpine catchment, monitored at event scale. The authors presented an exploratory analysis of the hydro-chemical composition of potential water sources and streamflow. They applied widely used, though not so novel, methodologies (simple hydrograph separation and EMMA), but complemented the analysis with hydrological connectivity simulations that make this study interesting. The work is well written, clearly structured and personally enjoyed reading it. Despite the short monitoring period, I find it with potential for publication in HESS after addressing a few suggestions.

We are happy to hear that you enjoyed reading our manuscript.

The concept of baseflow depends on the method used to estimate it and does not always describe active groundwater flow pathways. I suggest the authors describe what they defined in this study as baseflow

We agree that a definition of baseflow can be useful and will include a definition in the introduction. The definition will be along the following lines: "We define baseflow as the streamflow between storms and runoff events and assume that it comes from groundwater".

The third objective could be modified, it is well known that baseflow and rain mixture (negligible contribution of soil water) does not explain the changes in solutes concentrations in the streamflow.

We agree that the third objective should be modified and we will remove the redundant part of the question. We propose to change the third research questions as follows:

""In how far does conservative mixing of baseflow and rainfall explain the changes in stream solute concentrations and do discrepancies from this mixing indicate when other sources become connected to the stream?"

One of the principles of EMMA is that it relies on conservative tracers (not involved in adsorption or biological processes) and linear mixing process (Hooper, 2001). Did you analyse the conservative behaviour of the tracers? Please include the tests and state what tracers were used. Also, a graph showing the spatial-temporal concentrations of tracers in water sources would help the reader to contextualize their interaction during events.

Following the methodology of Barthold et al. (2011), we considered a tracer conservative when the concentrations are linearly correlated to at least one other tracer. We performed a linear regression on all streamwater, soil water and groundwater samples used in this study (n=549), and tested the correlations of Ca, Mg, Ba, Na, Fe, Mn, Cu, Zn, K, Cl, $D_{ex}$, $\delta^2H$ and $\delta^{18}O$. We set the threshold for a linear trend to $R^2 \geq 0.5$ and a p-value < 0.01 (i.e. the threshold of Barthold et al. (2011)). We found that EC, Ca, Mg, Ba, $\delta^2H$ and $\delta^{18}O$ are conservative, and that the other tracers were not. We would like to point out that we list most of these tracers as non-conservative solutes in the introduction (L57-58). If we use only the streamwater samples all tracers exhibit conservative behavior, except Mn and $D_{ex}$.

We understand the concern of the reviewer with regard to solving a non-linear mixing problem with a linear mixing solution. To avoid this issue we will reduce the tracer set used in the EMMA to EC, Ca, Ba, Mg, $\delta^2H$ and $\delta^{18}O$. We included an updated version of the figure and table that summarize the EMMA results using this tracer set (Figure 7 and Table 4 in the original manuscript). The largest

changes in the event-average fractions were for Event III, for which the estimated soil water contribution reduced from 0.38 to 0.01, and event II, for which the fraction of groundwater increased from 0.21 to 0.46 and the fraction of rainfall decreased from 0.45 to 0.29. The results for event I and IV changed only slightly.

[Figure]

Figure S2: PCA results and mixing diagrams for event I (top row) and event III (bottom row). Event I is representative of a small event, whereas event III is representative of an intermediately sized event. In the biplots (first column), the length of the arrow represents the explanatory power. The mixing diagrams based on the first two principle components (middle column) shows the individual rainfall (blue triangles), soil water (brown reversed triangles), and groundwater samples (black circles, red squares, green diamonds and black crosses, representing groundwater types 1-4 based on Kiewiet et al., 2019). The streamflow samples are shown with yellow (start of the event) to red (end of the event) triangles, and the average and standard deviation for each component is indicated with error bars. The third column shows a zoom in of the streamflow samples and highlights the evolution of the streamwater composition during the event (yellow = start, red = end), and the general direction of change indicated with a grey arrow and dashed lines.

Table S2. Event-average fractions of groundwater ($f_{GW}$), soil water ($f_{SW}$), and rain water ($f_P$) based on the three-component End-Member Mixing Analyses, and the associated uncertainties.

| Event | Three-component End-Member Mixing Analyses | | | |
|---|---|---|---|---|
| | $f_{GW}$ | $f_{SW}$ | $f_P$ | uncertainty |
| I | 0.81 | ~0 | 0.19 | 0.92 |
| II | 0.46 | 0.24 | 0.29 | 0.50 |
| III | 0.72 | 0.01 | 0.27 | 1.34 |
| IV | 0.73 | 0.02 | 0.25 | 0.91 |

Additionally, we will follow the suggestion of the reviewer to include a figure showing the spatial-temporal variability of tracer concentrations in each water source. We are testing different figure types but most likely it will be a figure that has a panel for each tracer and shows one boxplot per

source in each panel (Figure S1). We prefer to add this figure to the supplementary information but of course will mention it in the text.

Regarding EMMA's analysis, I suggest examining the evolution of events in the PCA space (Inamdar et al. (2013); Barthold et al. (2017); Correa et al. (2018)). Their dynamics and hysteresis can show the proximity of the streamflow to a certain source in the different stages of the event. Although as "soft data" it can bring insights into what groundwater or soil water contributes at a certain time.

We appreciate the suggestion of examining the evolution of events in the PCA space. We performed a similar analysis in an earlier stage of the data analysis but did not include the results in the final work. However, we re-examined the data and will add one figure per event to the supplementary materials. We plan include to a figure that shows the evolution of events in the PCA space by adding a panel to the figure showing the EMMA results (Figure 7 in the initial manuscript). We show this for event I and III in Figure S2, and for event II and IV in Figure S3.

[Figure]

Figure S3: PCA results and mixing diagrams for event II (top row) and event IV (bottom row). Event I is representative of a small event, whereas event III is representative of an intermediately sized event. In the biplots (first column), the length of the arrow represents the explanatory power. The mixing diagrams based on the first two principle components (middle column) shows the individual rainfall (blue triangles), soil water (brown reversed triangles), and groundwater samples (black circles, red squares, green diamonds and black crosses, representing groundwater types 1-4 based on Kiewiet et al., 2019). The streamflow samples are shown with yellow (start of the event) to red (end of the event) triangles, and the average and standard deviation for each component is indicated with error bars. The third column shows a zoom in of the streamflow samples and highlights the evolution of the streamwater composition during the event (yellow = start, red = end), and the general direction of change indicated with a grey arrow and dashed lines.

I am concerned about the very high uncertainties (Table 4), 160% in event III and 143% in event IV. Could it be due to the limited streamflow data, input-data uncertainty or time-dependent endmember variability (Chaves et al., 2008; Christophersen and Hooper, 1992). Unluckily end-member solutions do not exhibit low variability compared to the stream chemistry and not exhibit distinctive concentrations between end-members. I encourage the authors to analyse this limitation in more detail.

We agree that the uncertainties are extremely high and attribute this mostly to the high spatial variability in the tracer concentrations (i.e., input-data uncertainty). To be more explicit about this limitation, we will mention and quantify the other potential sources of uncertainty as well.

As an alternative the authors could refer to: Phillips, D. L. and Gregg, J. W.: Uncertainty in source partitioning using stable isotopes, Oecologia, 127(2), 171–179, doi:10.1007/s004420000578, 2001, to compute individual uncertainties in the calculation of source contributions to streamflow, this methodology considers the number of samples. The author could identify whether the uncertainties remain very high.

We appreciate the suggestion to investigate which of the water sources induces the largest uncertainty. The uncertainty analysis by Phillips et al. (2001) is very similar to the method that we used (Genereux, 1998). Both Genereux (1998) and Phillips et al. (2001) show that the uncertainty depends on the variability within each water source and in the mixture, and the differences in the composition of the water sources and the mixture. We will double check that these different methods give a similar uncertainty using the IsoSource mixing model (Phillips et al., 2005).

We agree that it is important that the reader is comfortable with the high uncertainties that we present, and thus that we need to elaborate the discussion of the high uncertainties. We plan to add such a description to discussion section 5.2 ("Which areas contribute to stormflow?").

The uncertainties that we present might be higher than for most other published mixing analyses because we use more groundwater and soil water samples than is typical. We have written a manuscript that exclusively deals with these high uncertainties. This manuscript is currently under review for WRR (soon to be resubmitted after minor revisions). If the WRR manuscript is accepted before the final publication of this paper, we will include a reference in the discussion, in addition to the revised discussion described above.

The introduction, methods and results sections are complete and clear to follow, despite some very long sentences that make a little difficult to follow the ideas.

We will carefully read through the text and split long sentences.

However, I find the manuscript poorly discussed. The authors support their findings in an extremely local context. The study would benefit from a broader perspective, comparing it with other similar ecosystems and/or with studies of the dynamics of water source contribution streamflow during events for example.

Upon re-reading the manuscript, we also recognize the weaknesses highlighted by the reviewer and the lack of a broader discussion and clear take-home messages. To overcome this, we plan to rewrite the discussion so that it also includes:

- A comparison of the results to literature from other study sites, as we did in the introduction. We think that, for instance, linkages to the studies of Ladouche et al. (2001) and Soulsby et al. (2007) would be useful here.

- A section that shows the broader impact of the study. We think that such a section should include how our results fit with current concepts of hydrologic connectivity (e.g., Blume and van Meerveld, 2015). It should also address the assumptions made with calculations of connectivity as mentioned above (e.g., Jackson et al., 2014).

In addition, we will rewrite parts of the existing discussion and emphasize the take-home message. We can achieve this with a section on the broader impact of our study as the final paragraph, and finish with a take-home statement. This could be something along the following lines: "The combination of hydrometric and hydrochemical data can be useful to identify hydrological connectivity and aid the interpretation catchment-scale runoff generation. However, we have to take the variability of the tracer concentrations in different water sources into account, as they can be large compared to the change in streamwater concentrations. The observed gradual deviations in the concentrations that are expected based on mixing of baseflow and precipitation are likely the result of increases in the contributions from many (small) landscape elements in the catchment and thus reflect the gradual increase in connectivity during events."

I assume the figures will be uploaded in a high-quality prior publication. In S1 please include rain and streamflow samples to visualize their distribution (potential streamflow at different colour scale for low, medium and high flows) and check the paper for a few typos.

Indeed, the quality of the figures deteriorated significantly when the file was converted into a .pdf. All figures, except figure 2, are vector images. Therefore, it should not be an issue to provide figures at the required DPI standards.

Including the rain and streamflow samples in S1 will make the figure too busy. However, we can add a second panel that shows the streamflow and the rainfall samples using the same axes as for the groundwater and soil water panel (Figure S3). By doing so, the figure will still be readable, and the rainfall and streamflow data are included.

We will certainly proof-read the manuscript carefully after making all the edits.

[Figure]

Figure S4: PCA results for all groundwater (n=335) and soil water samples (n=116) taken during the nine baseflow snapshot campaigns (Kiewiet et al., 2019), and all streamwater (n=100) and rainfall (n=47) samples taken during the four events used in this study. The mixing diagrams (left and middle panel) show the individual soil water samples (left-hand panel, brown reversed triangles) and groundwater samples (left-hand panel, black circles, red squares, green diamonds and black crosses, representing groundwater types 1-4 based on Kiewiet et al., 2019) for the first two principal components. The middle panel shows the rainfall samples (blue triangles) and streamwater samples taken during low flow (Q low, one baseflow sample) in yellow, during high flow (Q high) in red and during all other flow conditions in orange (Q mid). The error bars in both mixing diagrams indicate the average and standard deviation for each component (orange, brown and black error bars for streamwater, soil water and groundwater, respectively). In the biplot (right-hand panel) the length of the arrow represents the explanatory power for the solutes. The explanatory power of the first two principal components (PC1 and PC2) was 23.4 and 16.4%, respectively.

**References:**

Barthold, F. K., Tyralla, C., Schneider, K., Vaché, K. B., Frede, H.-G., and Breuer, L. ( 2011), How many tracers do we need for end member mixing analysis (EMMA)? A sensitivity analysis, *Water Resour. Res.*, 47, W08519, doi:10.1029/2011WR010604.

Blume T, van Meerveld HJI. 2015. From hillslope to stream: methods to investigate subsurface connectivity. Wiley Interdisciplinary Reviews: Water 2 (3): 177–198 DOI: 10.1002/wat2.1071

Genereux, D. ( 1998), Quantifying uncertainty in tracer-based hydrograph separations, *Water Resour. Res.*, 34( 4), 915– 919, doi:10.1029/98WR00010.

Jackson, C. R., Bitew, M., & Du, E. (2014). When interflow also percolates: Downslope travel distances and hillslope process zones. Hydrological Processes, 28(7), 3195–3200. https://doi.org/10.1002/hyp.10158

Ladouche B, Probst A, Viville D, Idir S, Baqué D, Loubet M, Probst J-L, Bariac T. 2001. Hydrograph separation using isotopic, chemical and hydrological approaches (Strengbach catchment, France). Journal of Hydrology 242 (3–4): 255–274 DOI: 10.1016/S0022-1694(00)00391-7

Soulsby, C., Tetzlaff, D., Van Den Bedem, D., Malcolm, I. A., Bacon, P. J., & Youngson, A. F. (2007). Inferring groundwater influences on surface water in montane catchments from hydrochemical surveys of springs and streamwaters. Journal of Hydrology, 333(2-4), 199-213. https://doi.org/10.1016/J.JHYDROL.2006.08.016

---

## Author Comment (AC3) · 3 Apr 2020

[Figure]

Figure S2: PCA results and mixing diagrams for event I (left) and event III (right). Event I is representative of a small event, whereas event III is representative of an intermediately sized event. In the biplots (first row), the length of the arrow represents the explanatory power. The mixing diagrams based on the first two principle components (middle row) show the individual rainfall (blue triangles), soil water (brown reversed triangles), and groundwater samples (black circles, red squares, green diamonds and black crosses, representing groundwater types 1-4 based on Kiewiet et al., 2019). The streamflow samples are shown with yellow (start of the event) to red (end of the event) triangles, and the average and standard deviation for each component is indicated with error bars. The third row shows a zoom in of the streamflow samples and highlights the evolution of the streamwater composition during the event (yellow = start, red = end), and the general direction of change is Indicated with a grey arrow and dashed lines.

[Figure]

Figure S3: PCA results and mixing diagrams for event II (left) and event IV (right). In the biplots (first row), the length of the arrow represents the explanatory power. The mixing diagrams based on the first two principle components (middle row) show the individual rainfall (blue triangles), soil water (brown reversed triangles), and groundwater samples (black circles, red squares, green diamonds and black crosses, representing groundwater types 1-4 based on Kiewiet et al., 2019). The streamflow samples are shown with yellow (start of the event) to red (end of the event) triangles, and the average and standard deviation for each component is indicated with error bars. The third row shows a zoom in of the streamflow samples and highlights the evolution of the streamwater composition during the event (yellow = start, red = end), and the general direction of change is Indicated with a grey arrow and dashed lines.

---

## Author Response (AR1)

We appreciate the helpful comments of the reviewers and editor. Please find below in blue font, the summary of how we have addressed each review comment in the revised manuscript. The line numbers in the responses refer to the revised manuscript.

Reviewer #1

This paper investigates spatio-temporal variability of end-member chemistry in a mountainous catchment. In a second step, EMMA is performed for four runoff events determining that soil sources contribute in addition to baseflow and precipitation, but groundwater being the dominating component. Additionally, the authors tested whether concentration of geochemicals could be calculated from conservative mixing. This was not the case. The authors also discussed the potential link between chemistry and changing hydrological connectivity. I find the study and the data set quite interesting. The paper is well written and data analysis is clearly described. While I like to overall paper good, there are several limitations.

We appreciate the overall positive assessment of our work and the helpful suggestions to improve the manuscript.

1. While I like the research questions and the introduction, I do not think that the research gaps for questions 1 and 2 are convincingly presented. For question 3 (first part), I believe that literature shows that this is not the case for most catchment where three component EMMA is performed.

We thank the reviewer for this comment. We recognize that the content and structure of the introduction did not logically lead to the research questions. We changed the introduction such that it provides more background for research questions. We changed the third research question as follows:

"How much do the changes in the concentrations of conservative and non-conservative tracers differ during events and does this difference provide information on the relative contributions of different parts of the catchment and, thus, hydrological connectivity?"

2. The connectivity part is a little bit weak. It is only loosely linked to the results and could be made stronger in results and discussion. The study also lacks a clear definition of hydrological connectivity. Is it mass transport here?

We rewrote the discussion to address this and other comments regarding the discussion. We included more linkages to the connectivity part (various locations in section 5.2 and 5.3), discuss the assumption that a connected water table indicates hydrologic connectivity (L295-300 and L525-531), and highlight points of attention for future connectivity studies (L408-410, L467-469, L531-532).

We agree that defining hydrological connectivity is important and that this will help the reader understand the study better. We now include a definition of hydrologic connectivity in the introduction (L41).

As connectivity here is linked to GW level rising close to the surface. Several recent papers challenged such a simplified assumption (e.g. Jackson et al., 2014, Klaus and Jackson, 2018, Gabrielli and McDonnell, 2020). I guess this is still somewhat in the debate, but clearly data on bedrock permeability should be presented to check whether the assumed connectivity from GW levels can be realistic. Maybe other proof can be provided that GW level can be used to infer connectivity?

To address this question, we now include the following information in the manuscript: "At most locations in the Studibach, there is an almost permanent water table in the low conductivity gleysols. We assume that significant lateral flow occurs when this water table rises into the near-surface layers, where the hydraulic conductivity is much larger (cf. Schneider et al., 2014). Hence, the

simulated connectivity refers to groundwater flow in the more permeable layer of the soil, above a saturated soil, and does not consider seepage to the bedrock" (L295-300). We also discuss this assumption and the potential effects of a smaller downslope travel distance due to bedrock seepage on our connectivity simulations (L525-531).

3. While I think that the paper is quite good, the discussion is currently weak. While the authors are discussing the data and their variation in detail (which is appreciated), I miss discussion of the broader impact of the study, as well as a better link to the introduction or the literature in general. Right now, the discussion refers to only a few studies, mainly related to processes in the same catchments. The authors need to present the broader implication of their work, and make their general contribution to the state of the art outside their study site clearer. At the end of the read I was a little unsure on the take home message. I really think the impact of the paper would be much better if that is achieved.

To address the weaknesses indicated by the reviewer, we expanded the discussion and added comparisons to various other studies and study sites (section 5.2 and 5.3). We also addressed the assumptions made to calculate the hydrologically connected area (section 5.3), included a section describing the influence of soil water (section 5.2) and expanded the link to interpretations of hydrologic connectivity (section 5.2 and 5.3). We also added some text to emphasize the take-home messages in the discussion and conclusions.

4. The majority of the figures need to be reworked (3, 5, 6, 8). They lack the quality that is needed for publication.

We revised the color scheme and style of our figures in the manuscript and supplementary material and enlarged all font sizes and points.

We will make sure to fulfil the required DPI when uploading the figures.

Minor comments:

L35: typo "McGuire"

changed

L47: The authors present catchment size and location for the Maimai; one could do the same for the Rietholzbach. The introduction generally good; the research gaps for the first two research question should be made more clear.

We now mention the catchment sizes and locations throughout the manuscript and adapted the introduction such that the research gaps are presented more clearly.

L92: Why should it only be baseflow? The literature is quite clear that, if tested, this is barely the case. So why asking a question we know to be not true?

We see that our research question could be rephrased to highlight the novelty of our work, rather than the findings from previous studies. As such, we changed the third research question to the following: "How much do the changes in the concentrations of conservative and non-conservative tracers differ during events and does this difference provide information on the relative contributions of different parts of the catchment and, thus, hydrological connectivity?"

L150: That is a valid assumption; but how variable is soil water chemistry (yes, the data is partly presented, but it could be stated)? Additional some more information on the choice of geochemicals and their commonly observed behaviour would be nice.

To address this comment, we included more information on the choice of the solutes (L235-242) and their behaviour in the catchments (L277-284). We stated that the variability in each water source was large (L274-276) and included a figure showing the variability of the various solute concentrations and isotopic compositions in rainfall, groundwater, soil water and streamwater in the supplementary material (S1 in this document, S2 in the manuscript) and refer to this figure in the text (L275 and L378).

[Figure]

**S1: Boxplots of the tracer concentrations for the different water types: groundwater (G), rainfall (P), streamflow (Q) and soil water (S). Each boxplot contains all streamflow or rainfall samples taken during the four events or all soil water or groundwater samples taken during the snapshot campaigns used in the study. Units for the isotope tracers are ‰ and for chemical tracers µg L$^{-1}$. Please note that y-axes differ for each panel, and that the y-axes of the panels on the bottom two rows are logarithmic for better visual comparison.**

L188: A clear definition of connectivity is needed, especially when not investigating the mass flux directly.

We agree that adding a definition of connectivity is helpful, and did this in the introduction (L41).

L198/199: You can only assume connectivity in cases where one have a low permeable of underlying bedrock (cf. Jackson et al., 2014; Klaus et al., 2018; Gabrielli and McDonnell, 2020).

We realize that we did not explicitly mention or comment on this assumption and now address it in the methods (L295-300) and the discussion (L525-531).

L219: Define "similar"

The difference in the event water fraction for the two-component hydrograph separation using $\delta^2H$ or $\delta^{18}O$ as a tracer was 0.05. We now define this in the methods (L227)

L251ff: There is a nice paper by Harris et al. (1995) that looked into changing end-member contributions. The idea is not too different from the one here.

Thank you for this suggestion. Indeed the paper presents a framework that is interesting for our manuscript. We now mention this paper in the discussion (L538).

L251ff: There is a range of studies that looked (e.g. McCallum et al., 2010), related to hydrograph separation, how GW chemistry is different from baseflow chemistry.

Thank you for recommending this paper. We found that the McCallum et al. (2010)-paper is very interesting, but that its focus on mixing in river-banks does not fit our revised manuscript so well. We found that another paper by the same author (McCallum et al., 2012) fits better to our revised manuscript, and referred to this paper in the discussion (L422-426). We do discuss the influence of mixing processes on the composition of groundwater contributions to the stream. To this end, we refer to the Chanat and Hornberger (2003)-paper, since it is focused more on hillslope-riparian zone mixing (L532-536).

L345: Or does that indicate a much less pronounced connectivity compared to the model?

We appreciate this suggestion and recognize that indeed, a less pronounced connectivity change might also be a valid reason for the smaller change in streamflow composition than expected. We now include this alternative interpretation in the manuscript (L524-525) and also added a more general comparison of the model results and observations to the discussion (L510-540).

L365: Is that surprising? The spatial variability is the maximum extend of the mixing diagram of endmembers. Thus, changes in the stream must be smaller, if the sampling was representative. I am missing the bigger picture here. The discussion is very detailed and evolves around the data being non-conclusive. It would be nice to expand this section and discuss what the key contribution to the field of runoff generation is. How do you go beyond studying this catchment? How does your work related to previous work? What is the key novelty? You may also think of linking your discussion better to the introduction and the used references there.

Indeed, this is not surprising, but very few studies have characterized the spatial variability for groundwater and soil water. We now more clearly mention that this finding can be expected (L398-401), and should be investigated at other sites as well (L408-410).

We addressed the second part of this comment by expanding our discussion with comparisons to the other studies and study sites that were mentioned in the introduction (section 5.2 and 5.3) and including points of attention for future connectivity studies (L408-410, L467-469, L531-532).

L448: but for some? And what do you infer from that?

Indeed, the contribution of soil water is important during some of the events, but not all events. We now discuss the implications of soil water contributions to streamflow in the description of the hydrologic functioning and expanded the discussion on the importance of soil water for hydrologic connectivity studies (L452-469).

Figures 3, 5, 6, 8 are not very well done. While the content is fine, the presentation, choice of colours, font size, and point type should be revised.

Thank you for pointing this out. We adapted our figures as described above.

**References;**

Chanat, J. G. and Hornberger, G. M., Modeling catchment-scale mixing in the near-stream zone—Implications for chemical and isotopic hydrograph separation, Geophys. Res. Lett., 30, 1091, https://doi.org/10.1029/2002GL016265, 2003.

McCallum, J. L., Cook, P. G., Brunner, P., Berhane, D., Rumpf, C., and McMahon, G. A., Quantifying groundwater flows to streams using differential flow gaugings and water chemistry, J. Hydrol., 416-17, 118-132, https://doi.org/10.1016/j.jhydrol.2011.11.040, 2012.

Reviewer #2

The manuscript entitled "Do streamwater solute cocentrations reflect when connectivity occurs in a small pre-alpine headwater catchment?" by Leonie Kiewiet, Ilja van Meerveld, Manfred Stähli and Jan Seibert, presents an important contribution to the understanding of the hydrological connectivity (or non-connectivity) processes that occur in a pre-alpine catchment, monitored at event scale. The authors presented an exploratory analysis of the hydro-chemical composition of potential water sources and streamflow. They applied widely used, though not so novel, methodologies (simple hydrograph separation and EMMA), but complemented the analysis with hydrological connectivity simulations that make this study interesting. The work is well written, clearly structured and personally enjoyed reading it. Despite the short monitoring period, I find it with potential for publication in HESS after addressing a few suggestions.

We are happy to hear that the reviewer enjoyed reading our manuscript.

The concept of baseflow depends on the method used to estimate it and does not always describe active groundwater flow pathways. I suggest the authors describe what they defined in this study as baseflow

We agree that a definition of baseflow is a useful addition and included the following definition in the methods: "We define baseflow as the streamflow between rainfall-runoff events, and assume that it comes from groundwater" (202-203).

The third objective could be modified, it is well known that baseflow and rain mixture (negligible contribution of soil water) does not explain the changes in solutes concentrations in the streamflow.

We agree that the third question should be modified and changed it as follows:

"How much do the changes in the concentrations of conservative and non-conservative tracers differ during events and does this difference provide information on the relative contributions of different parts of the catchment and, thus, hydrological connectivity?"

One of the principles of EMMA is that it relies on conservative tracers (not involved in adsorption or biological processes) and linear mixing process (Hooper, 2001). Did you analyse the conservative

behaviour of the tracers? Please include the tests and state what tracers were used. Also, a graph showing the spatial-temporal concentrations of tracers in water sources would help the reader to contextualize their interaction during events.

We reduced the tracers used in the EMMA so that it only includes conservative tracers. We tested for each tracer if the response was conservative based on the method of Barthold et al. (2011), and describe the test results in the methods (L246-243).

We also added a figure showing the variability in tracer concentrations in the different water sources and streamflow in the supplementary material (S1 in this document, S2 in the manuscript).

Regarding EMMA's analysis, I suggest examining the evolution of events in the PCA space (Inamdar et al. (2013); Barthold et al. (2017); Correa et al. (2018)). Their dynamics and hysteresis can show the proximity of the streamflow to a certain source in the different stages of the event. Although as "soft data" it can bring insights into what groundwater or soil water contributes at a certain time.

We appreciate the suggestion of examining the evolution of events in the PCA space. We now added a panel to Figure 7 and S4 of the revised manuscript that shows the evolution of streamflow during events in the PCA space. From these figures, it is also clear that in the composition of the streamwater samples is very close to the composition of the groundwater samples.

I am concerned about the very high uncertainties (Table 4), 160% in event III and 143% in event IV. Could it be due to the limited streamflow data, input-data uncertainty or time-dependent endmember variability (Chaves et al., 2008; Christophersen and Hooper, 1992). Unluckily end-member solutions do not exhibit low variability compared to the stream chemistry and not exhibit distinctive concentrations between end-members. I encourage the authors to analyse this limitation in more detail. As an alternative the authors could refer to: Phillips, D. L. and Gregg, J. W.: Uncertainty in source partitioning using stable isotopes, Oecologia, 127(2), 171–179, doi:10.1007/s004420000578, 2001, to compute individual uncertainties in the calculation of source contributions to streamflow, this methodology considers the number of samples. The author could identify whether the uncertainties remain very high.

We investigated the high uncertainties that we reported in the previous version of the manuscript. We found a mistake in our uncertainty calculation based on the Genereux (1998) method. We forgot to square a denominator term in the equation. We redid and double-checked all calculations, and the uncertainties are now much lower. We also compared our new results with the uncertainty calculated using the IsoSource mixing model. The uncertainty estimations were within 0.03 of each other, and thus very similar.

We now also calculated the contribution of each source to the total uncertainty. This highlights that most of the uncertainty is due to the uncertainty in the groundwater contributions. We also found that the uncertainty due to the soil water contributions is higher for event II, the only event with considerable soil water contributions than in the other events. We describe these findings in the results (L364-368).

The introduction, methods and results sections are complete and clear to follow, despite some very long sentences that make a little difficult to follow the ideas.

We carefully read through the text and split long sentences.

However, I find the manuscript poorly discussed. The authors support their findings in an extremely local context. The study would benefit from a broader perspective, comparing it with other similar ecosystems and/or with studies of the dynamics of water source contribution streamflow during events for example.

We think that broadening the perspective was indeed helpful. To address this point, we expanded the discussion and added comparisons to other studies and study sites in section 5.2 and 5.3. These cover a range of different ecosystems, some similar, some different to ours.

I assume the figures will be uploaded in a high-quality prior publication. In S1 please include rain and streamflow samples to visualize their distribution (potential streamflow at different colour scale for low, medium and high flows) and check the paper for a few typos.

Indeed, the quality of the figures deteriorated significantly when the file was converted to a .pdf. We will make sure that the final figures are available at the appropriate quality.

We included the rain and streamflow samples in an updated version of supplement S1, and used a different scale for low medium and high flows.

[revised manuscript text omitted]
_{pe}$$f_{GW}$ | $f_{SW}$ | $f_{GW}$$f_P$ | $f_{SW}$ | $f_P$$f_{pe}$ | uncertainty |
| I | 0.69 | ~0. | 0.31 | ~0. | 0.91 | 0.16 |
| II | 0.21 | 0.33 | 0.45 | 0.60 | 0.76 | 0.31 |

| III | 0.8139 | 0.6938 | 0.7222 | 0.011.59 | 0.2778 | 0.1635 |
| IV | 0.7872 | ~0.25 | 0.7428 | 0.011.43 | 0.2597 | 0.1419 |